# Neural fatigue by passive induction: repeated stimulus exposure results in cognitive fatigue and altered representations in task-relevant networks

Stefano Ioannucci [1,2 ✉], Valentine Chirokoff[1,3], Bixente Dilharreguy[1], Valéry Ozenne[4], Sandra Chanraud[1,3] & Alexandre Zénon [1]

Cognitive fatigue is defined by a reduced capacity to perform mental tasks. Despite its pervasiveness, the underlying neural mechanisms remain elusive. Specifically, it is unclear whether prolonged effort affects performance through alterations in over-worked task-relevant neuronal assemblies. Our paradigm based on repeated passive visual stimulation discerns fatigue effects from the influence of motivation, skill and boredom. We induced performance loss and observed parallel alterations in the neural blueprint of the task, by mirroring behavioral performance with multivariate neuroimaging techniques (MVPA) that afford a subject-specific approach. Crucially, functional areas that responded the most to repeated stimulation were also the most affected. Finally, univariate analysis revealed clusters displaying significant disruption within the extrastriate visual cortex. In sum, here we show that repeated stimulation impacts the implicated brain areas' activity and causes tangible behavioral repercussions, providing evidence that cognitive fatigue can result from local, functional, disruptions in the neural signal induced by protracted recruitment.

[1] Institut de Neurosciences Cognitives et Intégratives d'Aquitaine (INCIA)—UMR 5287, CNRS, University of Bordeaux, Bordeaux, France. [2] Visual and Cognitive Neuroscience Lab, University of Fribourg, Fribourg, Switzerland. [3] École Pratique des Hautes Études (EPHE), PSL Research University, Paris, France. [4] Centre de Résonance Magnétique des Systèmes Biologiques, UMR 5536, CNRS/Université de Bordeaux, Bordeaux, France. ✉email: ste.ioannucci@gmail.com

Cognitive fatigue is characterized by a pervasive avoidance of mental effort, usually triggered by periods of sustained cognitive activity. It is a distinctively recognizable state that is often experienced on a daily basis: without fail, even the most energetic individuals ultimately succumb to fatigue and feel the urge to seek rest and recovery. Behavioral scientists have so far formalized two key aftermaths of cognitive fatigue. The first, objective, component is an incapacitation to carry out effortful mental actions with a quantifiable decline in performance[1]. The second, subjective, component consists of a sensation of mental exhaustion[1]. The methods of assessment of both components remain debated due to their murky overlap with other constructs, such as boredom and underlying levels of motivation, which are hardly controllable in experimental settings[1–3].

Despite its ubiquity, the causes of mental fatigue remain elusive but have been theorized along a functional and a motivational axis. The former focuses on posited alterations taking place in the circuitry directly subtending the effortful action due to overwork[4–7], while the latter stresses the importance of the drive to partake in actions according to their intrinsic reward and utility[3,8,9].

As the functional paradigm proposes that fatigue issues from accumulation of metabolites or depletion of resources over repeated recruitment of the same neural networks, this leads to the prediction that even passive neural stimulation should eventually produce fatigue in specific neuronal assemblies. In previous work, we provided support for this hypothesis by showing, across multiple experiments, that passive visual stimulation had tangible repercussions on a task that involved stimuli identical to the repeatedly presented ones[10].

These performance drops were mainly present, if not bound within, the portion of visual field that underwent continuous passive stimulation, observed only under conditions of higher arousal and cognitive load, which were induced by auditory tasks concurrent to the visual stimulation[10].

The striking specificity of the aforementioned findings provides evidence for fatigue-induced performance drops that are unequivocally dissociated from other confounds such as boredom, level of skill in the task, and motivation. Although the neural underpinnings of such behavioral phenomenon are unknown, its selectivity points to an impact on the cortical regions that are engaged by the stimulation, coherently with previous reports of fatigue-induced alterations in the task-relevant networks in humans[4,6,11,12] and rats[13]. Such specific disruption, caused by protracted activity, would strongly support the functional paradigm of fatigue and could help to shed light on its neural basis, which has yet to reach a clear consensus in the literature, as attested by the diverging reviews on the topic[8,14–17].

Therefore, in the present study, we set out to test if the decline in performance following sustained passive stimulation is related to altered cortical responses in the brain regions recruited by the fatiguing condition, a hypothesis for which we have so far only partial and indirect evidence[4,6,11,12].

To do so, we hindered visual processing in a specific region of space (quadrant) by repeatedly stimulating this region with flashing stimuli (saturation), to induce objective fatigue in the texture discrimination task (TDT)[18]. The task's goal at each trial is to identify the orientation (vertical or horizontal) of a peripheral target that may appear in either upper quadrant, while maintaining central fixation. Importantly, the task's difficulty was adapted to the level of skill of each individual. In-between TDT sessions, participants underwent saturation of all the possible targets in a single quadrant, hence allowing to probe the aftermath of prolonged exposure on performance.

Concerning the neural mapping of this effect, we hypothesized that each participant would have their own functional response to

the task-relevant stimuli, since cortical response to identical stimuli and tasks varies from individual to individual[19]. Therefore, we adopted a subject-specific functional approach, instead of a one-size-fits-all procedure, by employing multivariate pattern analysis (MVPA)[20] to run a classifier on the brain activity within the subject-specific clusters of voxels that responded to the target stimuli in each quadrant. These clusters were identified via a localizer task, and the classifier's performance was compared before and after saturation, within and between quadrants, similarly to human agents.

Univariate analysis of the pre-post brain scans was also performed to identify the cerebral regions where the alteration between sessions was most prominent across participants. We also expected the regions of the brain that were the most responsive to the stimulation to show the largest drops in encoding reliability, which we tested by correlating the voxel-wise change in classifier accuracy to the degree of response to passive stimulation.

Lastly, the evolution of self-reported fatigue and sleepiness across the experiment was monitored and correlations between classifier accuracies, brain activity estimates, and behavioral performance were assessed, along with correlations between objective and subjective measures of fatigue.

## Results

**Behavior.** The generalized linear mixed model run on the accuracy in the TDT revealed significant Session ($X^2_{(1,19200)} = 6.61$, $p = 0.01$) and Quadrant ($X^2_{(1,19200)} = 7.98$, $p = 0.005$) main effects, as well as a significant Session-Quadrant interaction ($X^2_{(1,19200)} = 23.61$, $p < 0.001$). Post hoc tests, corrected for multiple comparisons by Holm procedure, confirmed a lack of difference at baseline between the saturated and non-saturated quadrant ($\beta = 0.24$, CI $= -0.19$, 0.66, Z $= 1.4$, $p_{corrected} = 0.48$, $\exp(\beta) = 1.27$), and between baseline and conclusion for the non-saturated quadrant ($\beta = 0.13$, CI $= -0.21$, 0.48, Z $= 0.88$, $p_{corrected} = 0.76$, $\exp(\beta) = 1.14$), while highlighting a significant decrease between sessions in the saturated quadrant ($\beta = 0.64$, CI $= 0.02$, 1.27, Z $= 3.85$, $p_{corrected} < 0.001$, $\exp(\beta) = 1.91$; Fig. 1).

Thus, we successfully replicated the finding that ~40 min of passive visual stimulation hinder the performance in a behavioral task involving the same stimuli. Inducing specific, measurable, objective fatigue[10].

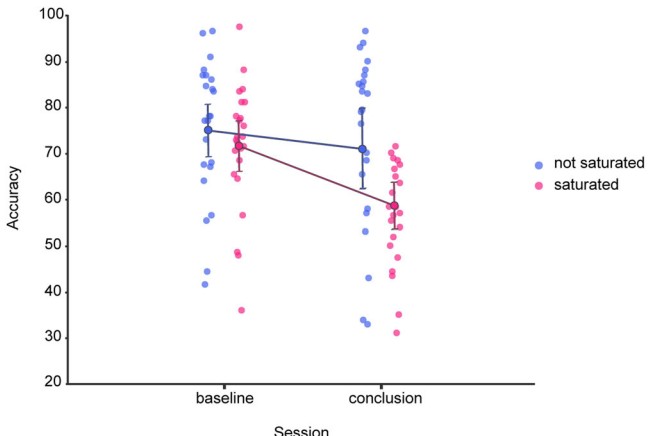

**Fig. 1 Behavioral results in the Texture Discrimination Task.** Graphical depiction of the average accuracy in the TDT (y-axis), between experimental sessions (x-axis) and condition (color). Error bars depict the 95% confidence intervals.

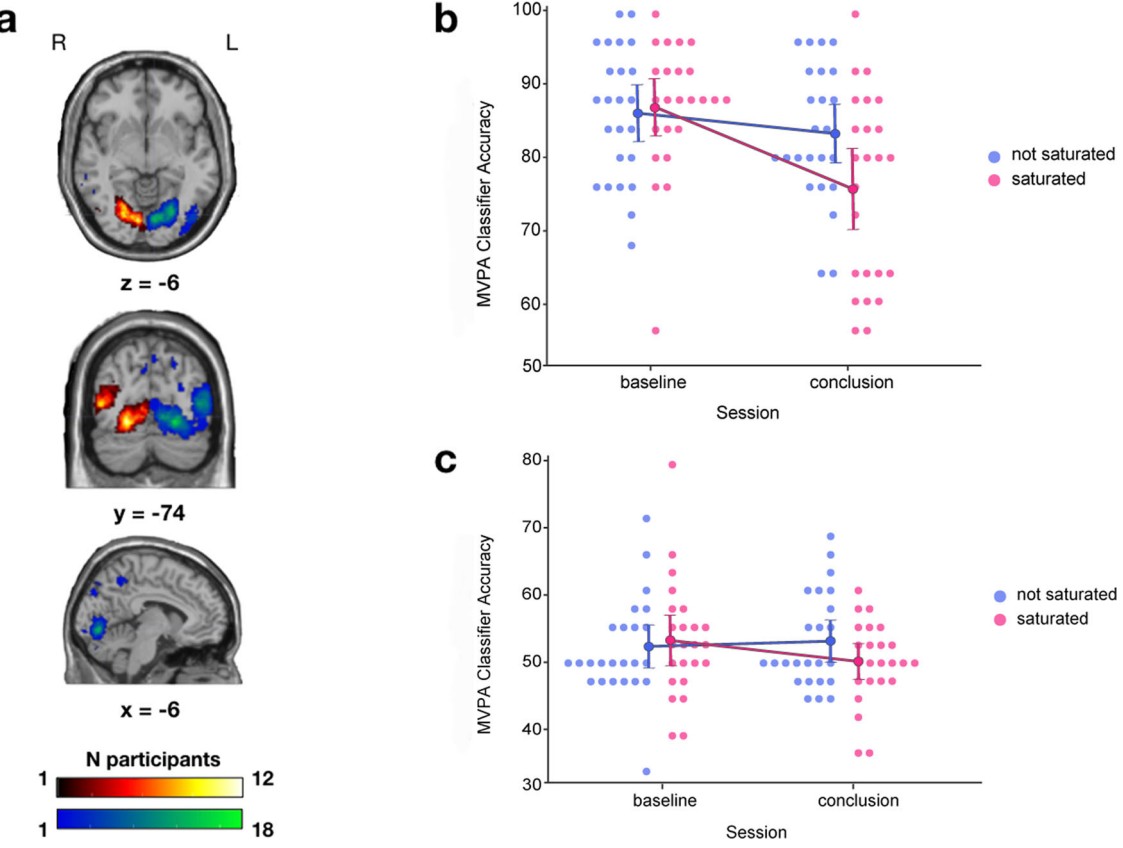

**Fig. 2 Aggregated subject-specific ROIs and MVPA results. a** Overlapping ROIs projected in the standard MNI single-subject T1-weighted MRI across participants, obtained from the union of voxels responding to stimuli in the left or right quadrant across pre and post localizer sessions, red-yellow = LQ, blue-green = RQ. **b** Graphical depiction of the accuracy of the classifier (y-axis) exploiting the brain activity of participants in response to stimuli during the localizer task, between experimental sessions (x-axis) for both experimental conditions (color). Error bars represent the 95% confidence intervals. **c** Graphical depiction of the accuracy of the classifier (y-axis) exploiting the brain activity of participants in response to stimuli during the TDT, between experimental sessions (x-axis) for both experimental conditions (color). Error bars represent the 95% confidence intervals.

**Subject-wise ROI determination.** Concerning the subject-specific regions of interest (ROI), we found via a paired samples t-test that significantly more voxels responded to stimulation in the right quadrant (RQ) than in the left quadrant (LQ) (mean 1889 vs 636, $p = 0.025$). However, this significant difference did not carry over when ROIs were divided into saturated and non-saturated (mean 1135 vs 1391, $p = 0.66$). In any case, an undeniable variability was found in the personal profile of response to stimulation, with the most overlapping areas lying within the visual cortex, common across at most 18 participants for the RQ and 12 for the LQ (Fig. 2a).

**MVPA on localizer brain activity.** The rANOVA on the classifier accuracy revealed the effect of Session ($F_{(1, 23)} = 12.59$, $p = 0.002$, $\eta^2_p = 0.35$) and the interaction of Quadrant and Session ($F_{(1, 23)} = 9.79$, $p = 0.005$, $\eta^2_p = 0.3$) as significant. A post hoc test on this interaction, corrected for multiple comparisons by Holm procedure, confirmed the absence of any baseline difference between saturation conditions ($T_{(23)} = 0.5$, $p_{corrected} = 0.67$), and yielded a significant difference in performance between baseline and conclusion, exclusive to the saturated quadrant (Fig. 4; $T_{(23)} = 4.43$, $p_{corrected} = 0.001$), while none was found in the non-saturated quadrant ($T_{(23)} = 1.25$, $p_{corrected} = 0.67$).

The MVPA findings thus display a remarkable consistency with the behavioral results, with a specific drop between experimental sessions in classifier accuracy, exclusively when exploiting the brain activity within the saturated ROIs (Fig. 2b).

**MVPA on TDT brain activity.** Regarding the assessment of the MVPA results employing brain activity during the TDT, the interaction between Quadrant and Session failed to reach statistical significance ($F_{(1, 23)} = 3.24$, $p = 0.085$, $\eta^2_p = 0.124$; Fig. 2c), along with any other effect.

**Univariate analysis.** The repeated-measures cluster-wise univariate analysis revealed two significant clusters showing significant decrease in response to the localizer task following saturation [$K = 139$, $K_z = 1.42$, $p_{fwe} = 0.017$; $K = 38$, $K_z = 0.97$, $p_{fwe} = 0.048$], on the side of the brain controlateral to saturation. These were anatomically located, by probabilistic cytoarchitectonic mapping via the Julich anatomy toolbox[21], in the lingual gyrus and inferior lateral occipital cortex, divided among functional regions V4 and V5 (Fig. 3a).

**Voxel-wise baseline activity in response to stimuli & change in classifying accuracy.** We next looked at the relation between the level of activation of individual voxels and their susceptibility to saturation-induced disruption. We hypothesized that voxels that responded the most to the stimulation would also be the most impacted by the saturation procedure. A mixed linear model highlighted the interaction of saturation condition and baseline beta activation as impacting significantly on the change in voxel-specific classifying accuracy, revealing that the voxels exhibiting the highest beta estimates at baseline in response to stimuli in the saturated quadrant also displayed the largest loss in classifying

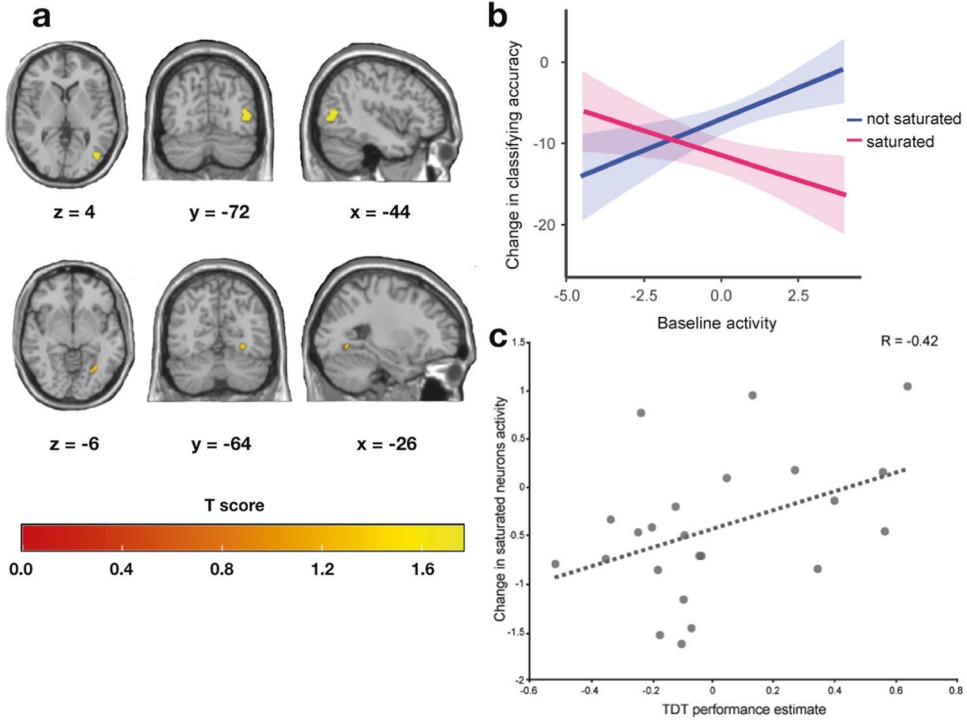

**Fig. 3 Univariate analysis results, relationship between baseline activity and classifying accuracy, brain–behavior correlation. a** Graphical representation of the clusters, controlateral to saturation, that presented significant changes in blood-oxygen-level-dependent signal before and after saturation across participants, projected in the standard MNI single-subject T1-weighted MRI. **b** Fit of the mixed model ran on the voxel-wise baseline activity in the subject-specific ROI (y axis) as a function of the change in the voxel-wise quadrant classifying accuracy (x axis) for both conditions (color). Shaded areas represent the standard error. **c** Scatterplot of the change in the estimated brain activity within the ROI responding to stimulation in the saturated portion of the visual field (y-axis) and the change in the performance of the behavioral task (x-axis) across the experiment, with the regression line depicted in dashed gray.

accuracy between experimental sessions, while the opposite was true when stimuli were present in the non-saturated quadrant ($\beta = -2.76$, CI $= -3.35$, $-2.17$, $T_{(36369)} = -9.15$, $p < 0.001$; Fig. 3b).

**Correlation between classifier performance and behavior**. No significant correlation was observed between the coefficients from the significant interaction in the behavioral analysis (Quadrant * Session) and the difference between baseline and conclusion accuracy of the classifier exploiting the saturated ROI in the localizer data ($R_{(21)} = -0.08$, CI $= -0.47$, 0.34, $p = 0.72$).

**Correlation between brain contrasts and behavior**. The correlation between the aforementioned estimates of performance in the TDT and the change in the univariate localizer brain signal within the subject-specific saturated ROIs revealed a significant positive correlation ($R_{(21)} = 0.42$, CI $= 0.01$, 0.71, $p = 0.048$; Fig. 3c), meaning that people who had a decrease in the average value of brain activity within the saturated ROI displayed a larger loss of performance in the TDT when the target was in the saturated quadrant.

**Subjective fatigue**. Concerning the pen and paper questionnaires, the statistical tests confirmed that there was a strong increase in the perception of both fatigue ($T_{(24)} = -7.61$, $p < 0.001$, $d = 1.52$) and sleepiness ($Z = -4.18$, $p < 0.001$, $r = -0.87$) during the experiment.

**Correlations between subjective and objective fatigue**. No significant correlation was found between the performance in the

TDT and the evolution of subjective fatigue between sessions, whether employing the random coefficients for the interaction between Session and Quadrant ($R_{(22)} = -0.22$, CI $= -0.57$, 0.22, $p = 0.29$) or solely those of Session ($R_{(22)} = -0.30$, CI $= -0.63$, 0.11, $p = 0.15$).

On the other hand, a significant negative correlation was found between self-reported sleepiness change and behavioral estimates of TDT performance in the saturated condition ($R_{(21)} = -0.54$, CI $= -0.79$, $-0.16$, $p = 0.008$, Fig. 4). To confirm the robustness of this latter finding to extreme values in the data, we ran the same variables through a Spearman correlation, which yielded similar results ($R_{s(21)} = -0.52$, CI $= -0.77$, $-0.14$, $p = 0.01$). Therefore, participants who performed worse in the saturated quadrant in the second TDT session tended to report greater levels of perceived sleepiness at the end of the experiment.

**Post hoc power analyses**. We have run exploratory tests to evaluate the statistical power of the reported results.

In regard to the repeated measures ANOVA on the MVPA classifier accuracies, the results from G*Power[22] state that with our reported effect size and employed sample size, a statistical power of 90% was reached, indicating that a significant result could have been found with a sample of only 10 participants.

With respect to the univariate analysis, we calculated the average Cohen's $d$ measure of effect size in the significant clusters for the post–pre contrast, which yielded a value of $d = -0.52$. This is generally interpreted as a medium effect size[23], and matches the higher end of the distribution of fMRI effect sizes reported in the study on the topic from Poldrack and colleagues[24], as could have been expected since we recorded activity in response to visual stimulation, which is known to

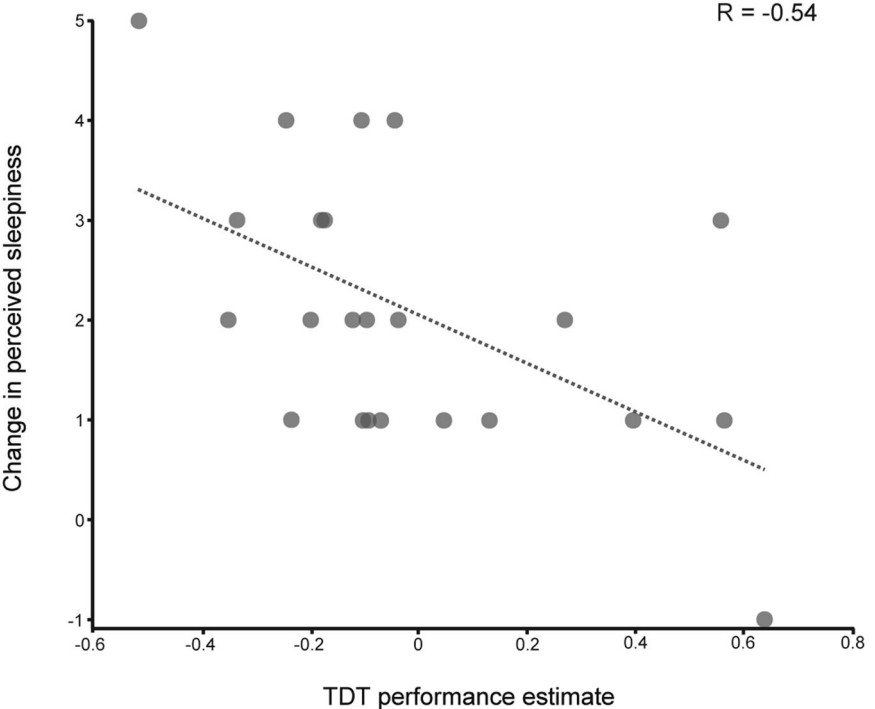

**Fig. 4 Correlation between reported sleepiness and behavioral performance.** Scatterplot of the change in the self-reported perceived sleepiness scores (y-axis) and the change in the performance of the behavioral task (x-axis) across the experiment with the regression line depicted in dashed gray.

display relatively large effect sizes compared to other manipulations. With our sample size, this translates in an achieved statistical power level of 68%, as calculated with G*Power.

Although reporting effect sizes should be standard practice in scientific endeavors involving statistical analyses[25], we would like to point out that post hoc power analyses should be approached with caution, as their use is a source of much debate within the scholarly community[26].

## Discussion
In the present work, we found evidence of specific, passively induced, neural fatigue. We replicated our finding that objective fatigue manifests itself after ~40 min of passive visual stimulation, in a behavioral task that involves identical stimuli in a delimited portion of visual field. This result has been found consistently across repeated experiments[10], when other variables such as motivation and level of skill in the fatiguing task are controlled for. Thereby, this objective fatigue effect cannot be explained by motivational accounts of cognitive fatigue[8,9], nor disproven by those that posit the objective component of fatigue as secondary with respect to the subjective one[3].

We also show that the accuracy of a classifier trained to decode participants' functional brain activity mirrored the results found in the behavior of those same participants, displaying a saturation-specific loss of performance (Figs. 1 and 2b).

Moreover, we found that voxels responding significantly to the stimuli employed in the saturation procedure exhibited the largest loss of classifying accuracy, suggesting a direct link between the level of activity during saturation and the consecutive disruption (Fig. 3b).

This discovery of specific neural fatigue was additionally corroborated by the correlation between the difference (conclusion—baseline; Fig. 3c) in measures of brain activity in response to the stimuli in the saturated quadrant within the saturated ROIs and the estimates of performance in the TDT. To our best knowledge, this set of results is one of the first cases of clear-cut relation

between purposefully induced neural fatigue and objective fatigue in human participants.

The above-mentioned behavioral consequences of fatigue occurred along with an increasing sense of tiredness and sleepiness in all the participants, as predicted. Admittedly, most experimental settings are likely to induce some measure of fatigue, sleepiness or boredom—all closely related constructs. However, the correlation we witnessed between the quantified performance alteration and the sensation of sleepiness (Fig. 4) provides a link between objective performance and subjective states that resonates with the notion that this construct is hardly distinguishable from fatigue[17,27,28], especially in experimental settings.

The present results indicate that by keeping specific neural assemblies active over a prolonged period of time, passive visual stimulation brings them to the point of disruption due to overwork[10], in agreement with recent findings that found decreased task-related responses in relation to fatigue development in the absence of alterations in the motivational circuit of the brain[6]. They also concur with research carried on habituation, neural adaptation and repetition suppression, which have thoroughly investigated the mechanisms by which relevant neuronal populations reduce their activity in response to repeated presentations of identical stimuli[29]. Indeed, the earlier theorizations of said phenomena explicitly framed them in terms of fatigue[30,31], and this view has been taken up more recently in a fatigue model that explains such effects by a reduction in firing rates caused by lessened synaptic efficiency, especially in the visual cortex[32]. A crucial feature of the present study is the performance drop that parallels the altered responses in the saturated portions of the brain. This is in stark contrast with standard neural adaptation, in which performance remains unaffected, or even improves, despite decreased neural responses, indicative of improved efficiency of encoding[33].

Our experiment was inspired by methodology employed in closely related scientific domains, such as perceptual learning[34–38]

and perceptual deterioration[11,39–41]. The majority of previous research stemming from these literatures has tended so far to theorize, or find, an involvement of the primary visual cortices in their phenomenon of interest. In the earliest reports on perceptual learning, the enhancement effect was interpreted in terms of local plasticity induced by retinal input at the level of orientation sensitive cells in the primary visual cortex[34].

Across the years, other investigators have built upon these theories by means of functional neuroimaging. For example, Schwartz et al.[42] found evidence for diverging activity within the lingual gyrus during TDT in humans, in a trained eye vs untrained eye contrast. Another study on the topic that did not focus exclusively on the early visual areas investigated the activity within the middle frontal gyrus, the superior parietal gyrus and the intraparietal sulcus, however, being limited to univariate contrasts of activity estimates within these a priori anatomical ROIs[43]. Studies on primates recording directly in the visual cortices have found extrastriate V4 neurons to be involved to a greater extent in perceptual learning than V1 cells[44,45].

Concerning perceptual deterioration, Mednick et al.[11] found proof of alterations in the signal within early visual area V1 following extended repetitions of the TDT (4 h in one day). Yet, also in this case, the followed procedure was to single out areas of the participant's brains and contrast the activity in each area with itself across experimental conditions.

Here, by combining the subject-specific approach with a univariate test between the brain scans at baseline and conclusion of the experiment, we were able to identify the cluster of voxels that were predominantly affected by the saturation procedure across the majority of participants, without enforcing an a priori anatomically determined comparison. As could have been expected, these clusters lied within the visual cortex, precisely within the lingual gyrus and the inferior lateral occipital cortex, functionally defined as areas V4 and V5.

Therefore, our results point to an involvement of the extrastriate cortex, in line with previous accounts in the domain of perceptual learning[42,44,45].

Indeed, V4 is thought to contain neuronal populations tuned to the orientation of basic visual stimuli[46], such as the lines employed in the present study, while V5 is believed to be involved in motion perception[47], making it presumably sensitive to their continuous 7.5 Hz fluttering.

Accounts of mental fatigue that posit its origin as the product of functional alterations that take place in over-worked cellular populations provide potential explanations as to which may be the processes taking place within these areas. One of the main candidates is the mechanism of consumption and storage of glucose, the brain's main source of energy. It has been argued that performance decrements arising during extended task repetition are related to abrupt use of this molecule, paired with reduced replenishing from the astrocyte support network[48,49]. It must be noted that the association between decreased global availability of glucose and cognitive activity has been contested by some authors as being confounded by extra-experimental factors[50]. Nevertheless, at the local level, research assessing the release of lactate, a molecule derived from glucose and thought to be the most immediately available fuel of neurons[51] found evidence of alterations in the concentrations of this metabolite, along with glutamate, in response to repeated visual stimulation related to concurrently measured blood-oxygen-level-dependent (BOLD) signal[52].

An alternative theory postulates the accumulation of metabolites as a consequence of prolonged cognitive effort[53,54]. Decreases in neuronal pH, presumably caused by lactate accumulation, have been shown to relate to decreased performance in a serial calculation task[55]. More recently, a study has found a relation between the behavioral manifestation of fatigue and increases in glutamate concentration in the task-relevant neural network[12]. Additional studies have associated glutamate[56], tryptophan[57], and tyrosine[58] with the subjective feeling of fatigue, however given the opacity of this construct further research is needed to validate such claims.

Finally, a last possible mechanism of cerebral functional alterations underlying fatigue may be local, use-dependent, sleep[59–61]. Accordingly, Vyazovskiy and colleagues[13] revealed, via deep electroencephalographic recordings, a remarkable correlation between specific neuronal assemblies displaying sleep-like activity and behavioral deficits in sleep deprived rats.

The accumulation of evidence for local use-dependent sleep has shifted the theory on sleep mechanisms from a static, global process, to a more dynamic and local process[7]. A similar shift may take place in the domain of cognitive fatigue, as this process is usually considered as a global mechanism, yet it may be well arising from the summation of several interconnected units that progressively reach the point of failure. This duality reflects, to some extent, the arguments between bottom-up functional accounts of fatigue and their top-down motivational counterparts. As is oft the case in science, the truth is likely to lay in the middle ground. One may suppose that the objective component of fatigue can be explained both by diminished cortical processing of the regions responsible for top-down motivational control, hindering the ability to partake in goal-directed behavior efficiently, with a concomitant decay of bottom-up signals in task-related regions due to metabolic constrains on prolonged activity. This view is compatible with recent frameworks put forth on the relationship between rising fatigue and its impact on the motivational circuits in the brain[62]. The subjective component would be a manifestation of these integrated processes, possibly with a distinct time-course[63], which would explain why they are rarely observed to correlate[1,64]. In such perspective, the two schools of thought of mental fatigue can be seen not as mutually exclusive, but explaining different aspects of this phenomenon, which is transversally recognized as multidimensional[1,2,15].

In any case, as mentioned in the introduction, the present study aimed at shedding light on where and if fatigue-induced functional alterations were occurring in the task-relevant neural networks, as found in other works employing different tasks, longer paradigms and univariate fMRI analyses[4,6,65]. Thereby, it was not designed to reveal the how, which will need careful, purposely designed, investigations.

The method employed here has proven reliable in inducing objective fatigue while controlling for common confounds, and may thus provide a reliable platform for further experiments. It is focused on the domain of vision, as this allows for relatively clean and straightforward experimental manipulations, thus affording the opportunity to probe the link between the overwork of specific neuronal assemblies and its behavioral, experiential, and neural consequences.

Our robust finding of objective fatigue in the visual domain is not trivial, as it is usually considered a domain where fatigue is all but absent (for example, see ref. [9]). This skepticism is likely explained by the fact that strict ocular fixation for prolonged periods of time is extremely rare, as eye movements and micro saccades prevent saturation to reach noticeable levels in everyday life. While in the present case neural fatigue was obtained from prolonged repetition of the same stimulus, we could expect the same effect to result from more ecological situations, as long as the same neural networks are solicited repeatedly.

For instance, we may consider the case of someone visiting a particularly vast and magnificent museum, such as the Louvre in Paris or the Vatican in Rome. Such person would surely have high levels of motivation in partaking in such activity, given the

substantial time and resources invested. Nevertheless, at a given moment during the visit, they may experience a progressively overwhelming sensation of fatigue, in response to the sheer amount of works of art and history that they are exposed to. In this case, while there certainly is no visual fatigue due to repetitions of identical visual stimuli, we may suppose that higher-order neuronal assemblies that process the numerous, complex images and abstract concepts and underlying associations that each exhibit implies eventually reach their point of saturation. This, in turn, prompts the motivational circuits of the agent to diminish the drive to keep engaged in the visit, urging him to seek a relieving change in activity—no matter the costs incurred or the genuine interest in the expositions itself.

**Limitations**. The main limitation of the present work is, presumably, the lack of reliable estimates of brain activity during the TDT, which transferred the burden of hypothesis testing onto the brain estimates of activity during the localizer sessions, as contemplated in the pre-registration (https://osf.io/vkrst).

In hindsight, this is likely explained by the diversity of the tasks. In the TDT trials, the stimuli are presented for very brief periods (0.34 s) and are mostly identical across conditions, as only the 3 peripheral target lines relocate from trial to trial, while the background embedding of horizontal lines remains unchanged. Accordingly, other experimental works seeking to investigate its neural signature have either devised methodological changes to compare between conditions and/or focused on predetermined anatomically defined areas[11,42,43]. On the other hand, localizer sessions had much longer, stable trials (12 s) with clearly distinct stimuli, thus allowing for more precise estimates of brain activity in response to them.

Our approach centered on task-related networks may have overlooked the contribution from task unrelated systems, and more specifically of facilitation and inhibition system, akin to those present in the domain of physical fatigue[16]. These systems were identified as the limbic-basal ganglia-thalamus-frontal network in the case of the facilitation system, and as the insular-posterior cingulate in the case of the inhibition system, with the latter found to be involved in fatigue on top of task-related regions only for pathological samples[16]. At any rate, the evidence supporting the existence of these systems is limited and comes from studies that do not involve purely perceptual tasks such as the present one[16].

No significant correlation was found between the accuracy of the MVPA classifier and the accuracy of participants. MVPA investigations that assess changes in activity inside brain regions in relation to the evolution of performance in a task are seemingly a minority in the literature, as most experiments are concerned with exploiting this method to localize brain networks underlying their phenomenon of interest. Within this minority, accounts of direct correlation between classifier accuracy and behavioral measures are scarce, with few notable exceptions[66,67]. As MVPA output is determined by the unique patterns of activation recorded rather than the magnitude of the activity, the overwhelming majority of brain–behavior correlations are carried out by taking the mean beta coefficient value in the relevant voxels, an approach with which we did find a meaningful correlation.

Likewise, we did not observe a significant correlation between subjective fatigue score change and objective fatigue performance decrement. Accordingly, such correlation is rarely observed in the field[1]. However, we had previously reported such a correlation employing similar methods in two separate samples[10]. In the present work, a correlation was found instead between the sleepiness reports and the behavioral performance, which may be partially explained by the supine position participants had to

assume during the MRI session[68], potentially causing the sleepy state to overshadow the fatigue state. In any case, sleepiness may be considered to be a request for rest in order to recover from fatigue, and the two often follow very similar evolutions[69–71]. Coherently, some contend that it is highly challenging to distinguish reliably between the two states[17,27,28].

Another point to be raised here is that beta-behavior correlation analyses in fMRI[72], and in general in the presence of small sample sizes[73] should be interpreted with caution. Replications will be necessary to conclusively validate the correlations highlighted in the present work.

**Conclusions**. Taken together, our results bring substantial proof that specific functional neural networks can be fatigued by overwork, even if passive. Further research is needed to characterize exactly what type of process, among candidates in the literature (glucose/lactate cycle, accumulation of metabolites such as glutamate or local sleep) takes place in the task-relevant functional brain networks.

We argue that the results found are a strong indicator that some forms of cognitive fatigue in healthy individuals stem from measurable changes in biological circuits caused by their repeated recruitment.

## Methods

The experimental design and main analyses performed in the present study were pre-registered on the Open Science Foundation platform (https://osf.io/vkrst).

**Design**. The experiment was carried out in two separate days (Fig. 5a). On the first day, participants would undergo a training on the TDT (Fig. 5b) to adapt the difficulty of the task to their level of skill, via a Bayesian staircase procedure[10], and to familiarize with the auditory tasks employed during saturation. On the following day, participants underwent the test session within the MRI scanner. Participants were first presented to a localizer task, in order to identify the portions of their brain which responded significantly to the stimuli, which were identical to the targets used in the TDT. Then, they carried out the TDT at their personalized difficulty. Following the first TDT session, participants underwent neuronal saturation (see below), which was then followed by a second TDT and localizer session. At the beginning and end of the test day, outside the scanner, participants were asked to fill out the Multidimensional Fatigue Inventory[74] and the Karolinska Sleepiness Scale[75] to assess the evolution in their perceived levels of fatigue and sleepiness during the experiment.

**Materials**. All experimental tasks were coded in Matlab 2019a (The MathWorks, Inc., Natick, Massachusetts, United States), using Psychtoolbox[76,77]. On training day, participants sat in front of a 1280 by 1024 computer screen at a distance of 60 cm and accommodated their heads on a chin-rest, using a keyboard to respond across the various tasks. On test day, participants lied in the MRI scanner and employed bimanual fiber optic response pads (Current Designs Inc., Philadelphia, USA) to respond during the TDT and auditory tasks. Sounds were presented in the scanner via OptoActive ANC headphones with active noise cancellation (Optoacoustics Ltd., Tel Aviv District, Israel). Heart and respiratory rate were recorded through the Physiologic ECG and Respiratory Unit of the Physiological Measurement Unit (Siemens AG, Bavaria, Germany).

On both days, an Eye-tracker (SR Research Ltd., Mississauga, Canada) was employed to ensure compliance of central fixation throughout the tasks, monitored by the experimenter.

**Texture discrimination task**. The main behavioral task consisted in the TDT, based on the task originally developed by Karni and Sagi[18]. The task's goal is to discriminate the orientation of a peripheral target, which consists of three diagonal lines aligned either vertically or horizontally, against a background of horizontally oriented bars (Fig. 5b).

Participants were instructed to maintain their gaze on a central fixation cross and report the perceived orientation of the peripheral target at the end of each trial, by pressing on the keyboard. These targets would relocate from trial to trial, with 2 possible locations, one for each quadrant of the upper half of the screen. Therefore, there were 4 possible targets: either horizontal or vertical, in either the left or right quadrant. In addition, to mitigate potential biases in the brain activity data, the response keys would invert in each block and their order alternated among participants, who were informed of this.

On training day, the task comprised 6 blocks with a balanced number of targets for each alignment (vertical/horizontal) and quadrant (left/right). The first 2 blocks

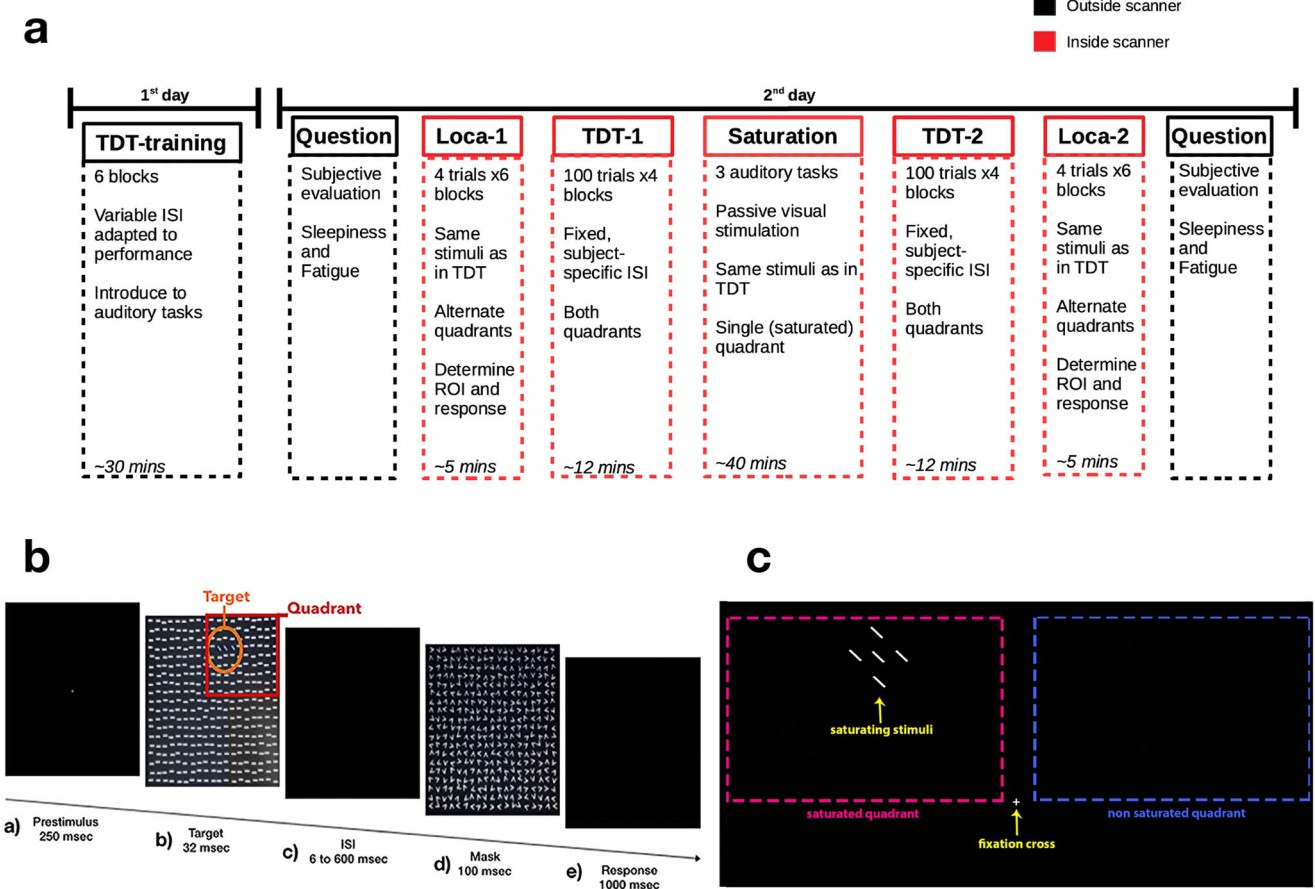

**Fig. 5 Schematic of experimental setup. a** Experimental design. **b** Breakdown of a Texture Discrimination Task trial. **c** Graphical depiction of the saturation procedure.

differed from the rest as they had a reduced number of trials (40) and extended interstimulus interval (ISI) durations and target display to allow the participants to familiarize with the routine. In the remaining 4 blocks, composed of 100 trials each, a Bayesian staircase procedure was used to determine the ISI depending on the participants' online performance, varying within the range of 0.02 to 0.6 s.

On test day, the subject-specific, fixed ISI corresponded to the value that had an 80% ratio of correct responses on the basis of the psychometric curve obtained from the block with the best performance in the training day. However, the participants' extracted ISI could not surpass a 0.6 s threshold, which was set as the maximum. Therefore, even if a participant's predicted ISI for an 80% correct performance was larger than this value, it would have been enforced to be 0.6 s. However, this wasn't necessary for any participant, as the average test-day ISI estimated from training across participants was of $0.22 \pm 0.13$ s, with the highest being 0.51 s.

To prevent the predictability of target onset given the fixed ISI, a random Inter Trial Interval was introduced at the beginning of each trial, varying within the range of 200 to 800 ms, uniformly distributed across trials and conditions.

A complete TDT test session consisted of 4 blocks, with equal numbers of trials per quadrant and target alignment, lasting ~15 min on average (~2 s per trial). Across participants, the order of target locations and alignments was predetermined to maximize the BOLD signal-to-noise ratio via design efficiency analysis (CANlab toolbox).

**Localizer.** To identify the areas involved in the processing of the task stimuli we devised a localizer task. Here, unique targets (e.g., horizontal alignment of three 45° lines in left quadrant, etc.) were presented in a block design to participants inside the scanner as they fixated the central cross.

Each unique target alignment and quadrant combination was presented 6 times for 12 s, alternating. The total duration of a localizer session was ~5 min.

Note that we opted to employ the actual target stimuli for this retinotopic mapping, rather than an unrelated grating checkerboard, as done in other works employing the same behavioral task[42,43].

**Saturation.** In between the baseline and conclusion TDT sessions, participants underwent a saturation session that lasted for 41 min. This saturation session

consisted in protracted visual stimulation, during which participants had to maintain their gaze on a central fixation point while the stimuli were continuously flashed fluttering at 7.5 Hz, with brief pauses between the end and the beginning of the auditory tasks in which the participants were engaged during saturation. These stimuli comprised all possible target line locations in one of the quadrants (i.e., the saturated quadrant, Fig. 5c), which was alternated between participants.

**Auditory tasks.** During saturation, participants were engaged in 3 different tasks that tax executive functions, short term memory and attention[10]. Specifically, we employed a 3-back task, a side task and a pitch-sequence task.

A trial in the 3-back task would consist of a list of 12 letters where participants had to report the occurrence of the target letter, which was any letter repeated 3 steps before in the sequence (3-back). A block was composed of 35 trials, each lasting 15 s.

In the side task, sounds were presented randomly either to the left or to the right earphone of the participant. These sounds came from different categories, namely: animal sounds or vehicle sounds.

Participants had a cue voice indicating, at random points during the block, which category they had to answer coherently to (i.e., if a sound of that category was presented to the left, they had to press the left key and vice-versa), this implicitly signaled they had to answer incoherently to the unmentioned category (i.e., if the sound was on the left, they had to press right and vice-versa). Furthermore, a third category of sounds was present, the computer/electronics category, to which they were instructed not to respond. A block would be made up of 135 sounds subdivided into trials of 5 sounds. A trial would last on average 18 s.

In the pitch-sequence task, participants were required to replicate a sequence of beeps that was presented to them. Four different beeps (from low-pitch (224 hz) to high-pitch (713hz) and in between (300 hz, 534 hz) were put together in randomly generated sequences. To each beep corresponded a key on the keyboard. These sequences comprised 4 to 6 beeps and a block consisted of 60 trials on average, with a mean duration of stimuli presentation of 3.6 s.

**MRI acquisition.** A 3T Magnetom Prisma scanner (Siemens, Erlangen, Germany) was employed using a 64-channel head coil. Subjects underwent anatomical and functional acquisitions during a single session. 3D T1-weighted MPRAGE

anatomical volumes were acquired with the following parameters: Repetition Time/ Echo Time = 2400/2.21 ms, Inversion Time = 1000 ms, Field-of-view = 256 mm$^2$, Matrix = 256 × 256 × 208, Slice Thickness = 1 mm.

All the functional MRI series were acquired using a 2D simultaneous multi-slice echo gradient echo planar sequence (2 × 2 mm voxels in-plane; 2 mm slice thickness with no gap; 40 transverse slices, 170 × 170 mm field-of-view; matrix 86 × 86; partial-fourier 6/8; repetition time = 1 s; echo time = 31 ms; multiband slice acceleration factor of 4; phase encoding direction Anterior-Posterior; flip angle 71°; bandwidth 1384 hz/pixel). The acquisition was aligned to the calcarine scissure of the participants.

**Quality control**. A quality control was conducted using mriqc software[78]. MRI images were visually inspected in order to remove those which presented major spatial artifacts (deformations and movements). In addition, fMRI series exhibiting a Framewise Displacement value greater than 2 mm or a DVARS value greater than 0.4% BOLD change were excluded for excessive motion according to Power's recommendations[79]. As mentioned, images from a single participant presented major deformations and were therefore rejected.

**fMRI preprocessing**. Susceptibility-induced distortions were estimated and corrected for all functional images using the top-up method[80] with FSL6.0.3 (FMRIB, Oxford, UK). In a second step, they were motion-corrected using linear transformation with the automated tool MCFLIRT in FSL and intensity-scaled to 1000. Then, the PhysIO Toolbox with MATLAB v9.7 was used to apply physiological noise correction[81]. Both cardiac and respiratory signal were recorded using a finger pulse oximeter and a pneumatic belt during MRI acquisition. Respective physiological regressors were created using the RETROICOR algorithm[82] as well as heart rate variability[83] and respiratory volume per time[84] and regressed out of from the functional MRI time series.

Due to the short repetition time used, no slice timing correction was applied. Finally, each series was realigned to the first localizer acquisition using rigid transformation with FSL flirt using mean as reference image and normalized to MNI space using SPM 12 by warping the T1 anatomical scans and applying the linear transformations to the functional scans.

**fMRI analysis**. During statistical contrasts, all the fMRI scans were high-pass filtered with a cutoff of 128 s/cycle, as per SPM default. For the scans used in the ROI determination contrast, an additional smoothing step was carried out at the default SPM value (8 × 8 × 8 mm), in order to more leniently determine the extent of the clusters, as voxels neighboring the peaks include potentially useful signal. This approach was preferred over directly enforcing a strict anatomical a priori region of interest, or choosing voxels from contrasts that aren't corrected for multiple comparisons which are expected to include more noise and thereby usually restricted a posteriori by some anatomical criterion (for an example see ref. [85]).

**Subject-wise ROI determination**. For each participant a contrast was carried out on their localizer data in both sessions (baseline and conclusion) between the targets in one quadrant versus the other (e.g., LQ > RQ) and vice-versa. Then, the clusters surviving Family Wise Error multiple comparison correction ($p_{fwe} < 0.05$) in both localizer sessions (AND operation) were combined into a single, subject-specific, binarized functional ROI, via the marsbar toolbox[86]. Therefore, for each participant we obtained two ROIs, one for each side of quadrant stimulation (Fig. 2a), which were later categorized as saturated or non-saturated, depending on which quadrant would display the stimuli during saturation.

**Multivariate pattern analysis**. MVPA was carried out both on the localizer beta coefficients and the TDT beta coefficients, with the same LQ vs RQ contrasts mentioned above as regressor. In the case of the localizer, the beta coefficients were derived by means of separate regressors for each single 12-s block of stimulation, while in the case of the TDT, regressors included groups of 10 successive same-condition trials[87].

MVPA analysis was carried out via The Decoding Toolbox[88], in the baseline and conclusion sessions, separately for the RQ and LQ ROIs. The classifier was asked, with a leave-one-out cross-validation procedure, to label in which quadrant were the stimuli presented, given the observed brain activity. Therefore, for each participant, we derived a data point of classifier accuracy for each quadrant, at baseline and conclusion, for both the localizer and the TDT brain scans. In addition, to estimate a voxel-wise measure of classifying accuracy, the above analysis was repeated with a searchlight procedure using a 4-voxel radius within the subject-specific ROIs. Searchlight analyses generate accuracy maps by measuring the variation in activity in multiple, overlapping, group of voxels in relation to the experimental conditions[89], thus outputting a measure of classifying accuracy for every voxel in the functional ROIs.

**Statistics and reproducibility**. Given previous results[10], and the current consensus on functional neuroimaging sample sizes[90], we pre-planned the study to have 24 participants. We eventually ran a total of 25 participants because one had

to be excluded from the fMRI analysis (due to visible artifacts in MR scans) and another from the behavioral analysis (due to a software failure to record his responses in the concluding session). We thus maintained a total sample of 24 in both the behavioral and neuroimaging analyses, except for the correlations as we had slightly differing samples: 24 for the MFI questionnaire (one missing participant), 22 for the Karolinska questionnaire (three missing participants), and 23 participants for the brain–behavior correlation (two missing participants).

Participants were recruited informally and took part voluntarily in the experiment ($M_{age} = 21.6 ± 1.8$, 10 m). All were naive to the experimental procedure, with no history of mental or visual conditions. Participants had normal or corrected-to-normal vision and provided written informed consent to participate. They received 50€ in compensation for their participation. The study was approved by the Ethical Review Board of the Comité de protection des personnes Sud-Est V.

rANOVAs on MVPA accuracies and behavioral analyses, except correlations which were done in Matlab, were carried out in jamovi[91] including the package gamlj[92] for linear models. Contrasts and analyses with neuroimaging data were carried out via SPM 12[93], unless otherwise specified.

Behavioral effects were assessed by means of a generalized linear mixed model on response accuracy in the TDT. Correct response in the task was modeled as a logistic dependent variable, with Session, Quadrant, and their interaction as explanatory variables, clustered by participant and also included in the random part of the model, with the following formula:

$$\text{Accuracy} \sim \text{Quadrant} + \text{Session} + \text{Quadrant} * \text{Session} \\ + (1 + \text{Quadrant} + \text{Session} + \text{Quadrant} * \text{Session}|\text{Subject}). \tag{1}$$

Based on previous results[10], we expected to find an interaction between Session and Quadrant, highlighting a specific drop in performance in the saturated quadrant between baseline and conclusion.

Changes in MVPA accuracy were evaluated by repeated measures ANOVAs with Session (baseline, conclusion) and Quadrant (saturated, not saturated) as within-subject factors, separately on the localizer and TDT estimates of brain activity. Here, we hypothesized that the effectiveness of the classifier exploiting the participants' neural signal would follow a similar pattern to their behavioral performance.

To identify precisely the spatial localization of the alteration induced by the saturation procedure, we proceeded to homogenize brain images in relation to their stimulation side (i.e., the brains of participants who were saturated on the left portion of the visual field were flipped, so that all participants had coherent saturated and non-saturated sides of the brain).

These beta maps were then contrasted with a classic second-level univariate approach, by means of a repeated-measures analysis with a non-parametric cluster-wise bootstrap procedure (SwE toolbox[94]), between conclusion and baseline scans of all the participants.

For this analysis, an explicit mask was restricted to the functionally relevant parieto-occipital portions of the brain, obtained by aggregating each subject-specific functional ROI and excluding any clusters anterior to the central gyrus and/or inferior to the lower bound of the occipital lobe.

Most voxels of the single ROIs fell within this mask (91% on average), with the majority of voxels located in the hemisphere contralateral to stimulation. The yielded data-driven mask is available for scrutiny in the public data repository. As a result of this analysis, we expected to find an alteration in the BOLD activations between baseline and conclusion of the experiment in the portions of the brain contralateral to the saturation procedure.

Classifier-behavior correlation (Pearson) was assessed by extracting the random coefficients of the Session-Quadrant interaction from the model on TDT accuracy data (representing individual deviations of each participant from the mean effect) and correlating these values to the delta (conclusion—baseline) in classifier accuracy in the saturated ROI of each participant, while brain–behavior correlation (Pearson) was assessed by correlating the same random coefficients to the delta (conclusion—baseline) in mean brain activity inside the saturated ROI of each participant. In this case, we expected to find a relation between the change in brain signal of the participants and their drop in task performance.

The quantification of voxel-wise deterioration in classifying accuracy in relation to the degree of response to passive stimulation was assessed by a linear mixed model on the delta in voxel-wise classifying accuracy with Baseline estimates of brain activity, Saturation and their interaction as explanatory variables, clustered by participant and included in the random part except for their interaction, as including interactions of between and within factors in the random part of a mixed model is not recommended[95], with the following formula:

$$\text{DeltaAccuracy} \sim \text{Quadrant} + \text{BaselineBeta} + \text{Quadrant} * \text{BaselineBeta} \\ + (1 + \text{Quadrant} + \text{BaselineBeta}|\text{Subject}) \tag{2}$$

This analysis was run to highlight the relationship between initial estimated activity in the relevant voxels and their change in classifying accuracy, as we hypothesized to see the most contributing voxels in the saturated portion of the brain to display the largest classifying accuracy loss.

The evolution of subjective fatigue was evaluated by a paired samples t-test on the average change towards fatigued responses in an adapted version of the Multidimensional Fatigue Inventory that included only the General Fatigue and

Mental Fatigue items[74], and by a sign-rank test between the baseline and conclusion scores in the Karolinska Sleepiness Scale[75]. In line with previous work[10], we expected to find increased sensations of fatigue and sleepiness following the experimental procedure.

The correlation between objective and subjective components of fatigue was assessed by correlating (Pearson) the random coefficients of the model on behavior and the delta (conclusion—baseline) in fatigue and sleepiness questionnaire scores of each participant. Similar to above, in light of our previous experiments[10], we expected to find a correlation between the subjective questionnaire scores and the estimates of behavioral performance.

All statistical tests were two-sided, where applicable.

**Reporting summary**. Further information on research design is available in the Nature Portfolio Reporting Summary linked to this article.

## Data availability

All the data employed in this work, along with instructions to reproduce the analyses, can be found at the following public repository: https://zenodo.org/record/7395703[96]. Moreover, the source data to recreate the plots present in the paper has been provided as supplementary data (Fig. 1—Supplementary Data 1, Fig. 2b—Supplementary Data 2, Fig. 2c—Supplementary Data 3, Fig. 3b—Supplementary Data 4, Fig. 3c—Supplementary Data 5, Fig. 4—Supplementary Data 6).

## Code availability

All the codes employed in this work, along with information on their purpose, may be downloaded from the following public repository: https://zenodo.org/record/756599[97].

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

## Acknowledgements

This research was funded by IdEx Bordeaux Junior Chair and Agence Nationale de la Recherche (ANR JCJC Cocogit 2019-0054).

## Author contributions

S.I. and A.Z. conceived and planned the experiment, S.I. carried out the experiment and statistical analyses with A.Z.'s supervision. V.C and B.D. contributed to the preprocessing of the fMRI data. The manuscript was written by S.I. in consultation with A.Z., S.C., and V.O.

## Competing interests

The authors declare no competing interests.
