## [Peer Review File · Communications Biology]

Reviewers' comments:

Reviewer #1 (Remarks to the Author):

The manuscript presents a study on the effects of repeated visual stimulus exposure on the development of local neural fatigue. Participants performed a texture discrimination task (TDT), before and after a fatigue-inducing saturation period. In the TDT visual texture stimuli (two different orientations) are presented in one of two spatial locations. Participants responded to the stimulus orientations with a button press. During the saturation period the same stimuli were being repeatedly presented, but only in one visual hemifield, presumably inducing fatigue in the contralateral visual cortex. This saturation procedure was passive (i.e. visual stimuli were presented without requirement to attend or respond to them, while an unrelated auditory task was performed), and lasted for 41 minutes. Results showed that detection performance after saturation deteriorated, but only for the stimulated hemifield. In addition, fMRI multivoxel pattern analysis (classifying the presented stimulus from BOLD activation patterns in identified stimulus-responsive areas) showed a drop in classification accuracy only for the saturated hemifield. This drop in MVPA classification performance was significant for activation patterns during the localizer scan, while a similar decline was not significant during the TDT task itself. Lastly, changes in BOLD activation were correlated with MVPA classifier performance, and with TDT behavioral performance, while subjective sleepiness was correlated with TDT behavioral performance.

I believe that this is a well-conceived and well-executed experiment that connects several inter-related, but theoretical perspectives that have so far existed rather isolated from each other. The TDT is a good choice of task for testing the hypothesis of passively induced neural fatigue, given its reliance on highly localized neural processing. The use of passive presentation to establish neural saturation is also an elegant way to avoid motivational factors usually intertwined with fatigue processes. From my reading, I believe that the data convincingly show that over-use of local neural pools can induce a) localized hypo-reactivity and degrade pattern representation, and B) performance decline confined to the saturated location. I mostly have some suggestions about the theoretical positioning of the paper.

There are two areas of research that have developed substantial thinking around the concept of (local) neural fatigue. The first is the area of local use-dependent sleep (Krueger et al. 2019; Siclari & Tononi 2017). The authors touch on this briefly in discussing the work by Vyazovzkiy et al. (2011). However, I believe that this area has produced a more elaborate body of work that may be of relevance to the current study. The idea that local neural ensembles temporarily go into an off-state due to over-use is highly similar to the concept of neural fatigue as advanced here. Both human (e.g. Huber et al. 2004) and animal work within this area, have inspired ideas about its mechanisms (e.g. neuronal sleep regulation through local inflammation), and functions (e.g. facilitating plasticity). In the latter aspect, this area also directly connects to the Texture Discrimination Task used here, which - as the authors indicate - had its origins perceptual learning, and sleep-dependent plasticity.

The second area that might be of relevance (esp in relation to the MVPA analysis) is the work on repetition suppression (i.e. reduced neural response to a repeated presentation of a stimulus). A prominent debate in this literature is whether repetition suppression reflects neural fatigue (an overall reduction in neural responsiveness to the stimulus), or neural sharpening (fine tuning of neural representational patterns). The current saturation paradigm could be seen as an extreme form of repetition suppression. Findings of reduced pattern classification in the current study might inform the debate on fatigue vs sharpening in the area of repetition suppression (for reviews see Grill-Spector et al. 2006; Barron et al. 2016).

Without forcing any obligation to veer too far away from the authors' original writing, I wonder if there is any merit to elaborating on (either or both) the above areas a bit more. I believe the current study would be in a good position to connect to these research areas, and both be informed by, and inform their respective theoretical debates.

Another comment about the theoretical framing is that the rationale is much built on contrasting the 'functional' and the 'motivational' accounts of fatigue. In the abstract they are presented as opposing views: "... it is unclear whether alterations due to prolonged mental effort take place in over-worked task-relevant neuronal assemblies, or in task unrelated networks involved in motivation regulation. Here, we tackle this question by means of a paradigm based on repeated, passive visual stimulation, ...". The introduction is a bit more cautious, saying that: "...the causes of mental fatigue remain elusive but have been theorized along a functional and a motivational axis". I follow the authors that the current design effectively isolates functional fatigue from motivational decline. At the same time, these effects need not be mutually exclusive, and could well occur simultaneously in most situations. I believe that a marked strength of the current study is that it probes neural fatigue, free motivational effects. However, I don't believe the study can arbitrate between functional and motivational accounts (as the abstract suggests).

Specific comments

- Regarding the behavioral correlations, I couldn't quite make up what the TDT performance estimate represented (and how to interpret it). From the Methods Statistical analyses it was described as the random coefficients of the GLMM, but it wasn't clear to me which coefficient that was (is it the Session x Quadrant interaction?). I wonder if it is possible to describe the model in formula, and indicate which coefficient is used to run the correlations.
- Page 3 paragraph 3: references seem not formatted. "(Blain, Mednick, Gergelyfi)"

References

Local use-dependent sleep

Krueger, J. M., Nguyen, J. T., Dykstra-Aiello, C. J., & Taishi, P. (2019). Local sleep. *Sleep medicine reviews*, 43, 14-21.

Huber, R., Felice Ghilardi, M., Massimini, M., & Tononi, G. (2004). Local sleep and learning. *Nature*, 430(6995), 78-81.

Siclari, F., & Tononi, G. (2017). Local aspects of sleep and wakefulness. *Current opinion in neurobiology*, 44, 222-227.

Repetition suppression

Barron, H. C., Garvert, M. M., & Behrens, T. E. (2016). Repetition suppression: a means to index neural representations using BOLD?. *Philosophical Transactions of the Royal Society B: Biological Sciences*, 371(1705), 20150355.

Grill-Spector, K., Henson, R., & Martin, A. (2006). Repetition and the brain: neural models of stimulus-specific effects. *Trends in cognitive sciences*, 10(1), 14-23.

Reviewer #2 (Remarks to the Author):

The authors of the study titled 'Neural fatigue by passive induction' set out to resolve the question of whether cognitive fatigue occurs due to depletion of neural resources (functional account) or changes in motivation to pursue rewards (motivational account). Predicting that passive stimulation of neural circuits involved in effortful activity would induce fatigue, they present a paradigm where two blocks of perceptual decision-making task are interspersed with a block of prolonged exposition ('saturation') to perceptual stimulus in one of the visual fields targeted in the decision-making task and compare performance, patterns of brain activation in individually-calibrated ROIs, mean activation in the ROIs and subjective reports of fatigue and sleepiness. The results show a drop in performance, a decrease in MVPA classifier accuracy and a decrease in mean brain activation in clusters in V4 and V5 specific to trials and ROIs targeted by the saturation procedure. They find that the drop in mean brain activity

(but not classifier performance) is correlated with the strength of effect on performance. The task is successful in eliciting higher levels of fatigue and sleepiness as measured with self-report questionnaires, but only the sleepiness increase is shown to correlate with a drop in performance. Authors conclude that passive stimulation of brain circuits induces neural fatigue specific to these circuits, mirrored in a drop in performance only in the task recruiting those and that these changes are responsible for the mental fatigue.

This is timely topic and an interesting study using a novel methodology to examine the still under-researched phenomenon of fatigue. Including links to the data and analysis scripts must be appreciated and commended. Although there are elements of the manuscript that are real strengths, the work does not provide compelling evidence in favour of their conclusions. Firstly, the initial theoretical tension (between motivation and neuronal fatigue accounts) as being opposed is overstated, the task cannot rule between such accounts, the sample size is relatively small, and many statistical results are not convincing. For these reasons the manuscripts falls somewhat short of conclusively showing neuronal fatigue in task-sensitive circuits is responsible for mental fatigue, which may leave it unable to make a significant impact on the field.

Major:

1. The main premise of the study is that accounts of fatigue either argue it is purely a motivational phenomena and other accounts suggesting it is based on neuronal fatigue. However, these accounts are not mutually exclusive and the dichotomy is somewhat overstated. More recent accounts all suggest that increases in neuronal fatigue might influence motivation through processes similar to meta-cognition (see Muller & Apps, 2019, the work of Steve Fleming or Meyniel et al., 2013). These accounts argue that costs accumulate in task-processing areas, and are monitored in other systems that then influence subsequent valuations of whether to keep working at the task (Pessiglione et al., 2018, Brain; Muller et al., 2021, Nature Comms). In this sense, task performance can depend on both processes. I.e. two people could have similar levels of neuronal fatigue but one person suffers from poor performance and another doesn't, because they can compensate by being more motivated. The problem for this manuscript, is that the whole paper is framed around showing effects are related to neuronal fatigue not motivation, but the second theory is never ruled out. As a result, much of the framing of the paper is over-stated and misleading.

2. Linked to this, conceptually, there are some issues with referring to performance drop as 'objective fatigue'. Fatigue is a complex and multi-faceted phenomenon, often accompanied by loss of motivation and a drop in performance, however, these objective measures are better seen as potential behavioural correlates of fatigue, which is an inherently subjective phenomenon, rather than its defining features. There are a number of problems with the definitions used throughout. The most problematic being that the study does not manage to show a relationship between the subjective measures of fatigue and behavioural and neural measures. As a result, much of the introduction and discussion about this work being about "fatigue" is misleading. Moreover, they only link task measures to subjective sleepiness and not subjective fatigue. This is a major problem for the framing of the paper is all around mental fatigue. Relatedly:

- a. Multidimensional Fatigue Inventory (once erroneously referred to as "Index" (p.13, please correct) was developed as an assessment of a persistent, pathological symptom of fatigue (developed specifically in cancer patients) rather than an evaluation of (non-pathological) fatigue that develops in response to exertion. With items like "I think I do a lot in a day" or "I tire easily" this inventory is not particularly suited to detect changes in effort-induced fatigue levels. I do not blame authors for this as a suitable measure has not, to my knowledge, been devised, however, it does lead to my questioning the justifiability of interpreting their results in terms of mental fatigue
- b. Following up from b), Multidimensional Fatigue Inventory is, by name and nature, multidimensional where authors purport to measure mental fatigue specifically
- c. Finally, the authors compare the subjective measure of fatigue change to 'objective measure' by the

means of correlating change in MFI score with the participant-specific coefficient in GLMM predicting the accuracy in the task. As far as I understand this is a coefficient quantifying interaction between task session (pre- vs post-passive stimulation session) and visual field (or 'Quadrant': saturated vs non-saturated), so the potential experimental manipulations that could explain the drop in performance. To fully explore this relationship it would be advisable to examine whether subjective fatigue is related to a drop in performance between sessions in and of itself, regardless of the saturation procedure. Although there was no significant difference between accuracy in two experimental sessions in non-saturated quadrant, based on the accuracy plot (Figure 5) and specifically the error bar for session 2 - non-saturated quadrant it appears that there were more substantial individual differences in participants' drop in performance in this condition.

3. During the saturation session participants were performing cognitively demanding auditory tasks. Very little is explained regarding the nature of these tasks, e.g., whether the effort level of these tasks was individually adjusted. It is likely that these tasks also had an effect on any changes in fatigue levels, so more information on these tasks would be welcome and on data on what participants were doing. This seems crucial to understand what "saturation" and fatigue effects actually mean in this task.

4. Overall, many of the key results that one would expect are not present (see links to subjective fatigue above), some results that are borderline or non-significant but interpreted as significant (see Fig.3c), the between-subject variability seems high in all key figures and in some analyses classifier accuracy is very close to 50% in all conditions. Given the small sample size, this leaves a concern that many of the results are not robust, and will likely not replicate. Of particular concern is that several analyses rely on correlations, which are known in small samples to be inflated (Yarkoni et al., 2009). Overall, even though I am sympathetic to the notion that interesting studies don't need large samples, this particular pattern of statistical results in a small sample appears concerning.

5. Linked to that point, is the fact that figure 5, appears to have 2 outliers in the perceived sleepiness scores top left and bottom right of the plot). Excluding those and reanalysing the data would be advisable to show whether or not these are driving the significance of the correlation.

6. In general, many aspects of the manuscript were a little unclear and lacking in important details (see each of the minor comments below). In particular, what tests were being conducted and how statistical results were derived was hard to follow and lacking the clarity required for reproducibility. Figure legends were lacking the detail required to understand what was being plotted

Minor:

7. Overall there is a general lack of reference to the most recent develops for research in mental fatigue. The only papers referenced after 2016 are to methodologically relevant papers, not conceptual or empirical on the topic at hand – fatigue.

8. The title could elaborate a little more on the contents of the study for greater clarity and impact. Passive induction is not a well parsed phrase or clear to people even in the field

9. As authors point out (in discussion, p.25), the approach of linking (or attempting to link) behavioural measures and MVPA classifier accuracy is not used often. It would therefore be useful to include a more elaborate rationale for this choice and underlying assumptions.

10. Please include the correlation methods used

11. In figure 2 please describe what the error bars depict

12. Since the results of staircase procedure were capped, please include information how many participants were capped at the maximum 0.6 ISI and how this affected their performance

13. Word-use on page 10 ('betas') and page 12 ('brains') should be more precise

14. Page 23: "Indeed, V4 is thought to contain neuronal populations tuned to the orientation of basic visual stimuli (Roe et al., 2012), such as the lines employed in the present study. While V5 is believed to be implicated, to some degree, in the perception of movement (Born & Bradley, 2005), therefore reasonably being expected to respond to the stimuli of the localizer trials, due to their continuous 7.5 Hz fluttering." - please correct the grammar
15. Page 25: "These systems were identified as the "limbic-basal ganglia-thalamus-frontal" network in the case of the facilitation system, and as the "insular-posterior cingulate" in the case of the inhibition system, with the latter found to be involved in fatigue on top of task-related regions only for pathological samples." - please include a reference
16. Page 25: "At any rate, the evidence supporting the existence of these systems is limited and comes from studies that do not involve perceptual tasks such as the present one." - please include references
17. Please ensure that acronyms (MVPA, TDT) are all capitalised in the subheadings

Reviewer #3 (Remarks to the Author):

In this manuscript, Ioannucci et al. used passive visual stimulation of a specific region of the visual field to induce a loss of performance specific to that region. They then used fMRI to find brain correlates of this effect, showing functional brain alterations in task-relevant networks. All in all, the paper is well-written, the experiment is cleverly devised and the results sound convincing albeit intriguing. I only have a couple of minor comments.

My main concern is about the interpretation of the results regarding the subjective feeling of (cognitive) fatigue. The beauty of the design is about the specificity of the effect. However, fatigue is a general feeling that seems to transfer from one task to the other (after a long session of reading/writing, I'm too tired to play chess!). Thus, even if I'm convinced by the results, I would be reluctant to call "cognitive fatigue" a decrease of performance specific to a very specific task. On a related topic, I would like the authors to elaborate on how they would imagine the link between (local) brain fatigue and what is reported by participants when they are required to rate their (global) level of fatigue. Is it supposed to be an average of the whole brain? Or maybe the degree of fatigue of the most "exhausted" region? Finally, my intuitive assumption would have been that vision (and its correlates) might be the most difficult function to fatigue... After all, we are receiving visual information all day long on the whole visual field every day... On the contrary, some functions (e.g. executive ones) seem particularly susceptible to fatigue. I understand why the authors chose this task given the methods they used, but as the paper is framed about cognitive fatigue, I believe they should elaborate about how their results could be extended to the cognitive fatigue experienced in everyday life.

The rationale of each analyze and expected results could be more explicitly detailed in the result section to facilitate the reading. This is particularly the case for negative results.

I'm not sure to understand the number of participants. I understand that one of the participants was excluded from fMRI analyses (and kept in behavioral ones) because of artefacts. However, since the most important analyses involve baseline and conclusion sessions, I'm not sure it makes sense to keep the participant that failed to respond in the concluding session...

Univariate analyses: By construction, the two clusters identified (Fig 4a) must be within the overlapping ROIs (Fig 3a) but it seems that at least the bigger one is not in a region common to most subjects. However, it is quite difficult to assess from eyeballing, would it be possible to provide an overlap of these two maps (flipping ROI when required).

Voxel-wise baseline activity in response to stimuli & change in classifying accuracy : If I'm correct, the

voxels that exhibited the lowest beta estimates at baseline in response to stimuli in the non-saturated quadrant displayed the largest loss (opposite of what was observed in the saturated one). I was quite surprised by this result (I would have expected a lack of relation). Could the authors elaborate on this finding?

Fig 4.C (axe and legend): I'm not sure speaking about "neurons" is appropriate here (speaking about the brain activity of neurons was quite funny though!)

Which random coefficients of the GLMM were correlated with questionnaire scores? According to the result section, I believe sleepiness was correlated to change of performance in the saturated quadrant but not in the non-saturated one?

Was there some correlation between questionnaires and fMRI findings?

Reviewer #1 (Remarks to the Author):

The manuscript presents a study on the effects of repeated visual stimulus exposure on the development of local neural fatigue. Participants performed a texture discrimination task (TDT), before and after a fatigue-inducing saturation period. In the TDT visual texture stimuli (two different orientations) are presented in one of two spatial locations. Participants responded to the stimulus orientations with a button press. During the saturation period the same stimuli were being repeatedly presented, but only in one visual hemifield, presumably inducing fatigue in the contralateral visual cortex. This saturation procedure was passive (i.e. visual stimuli were presented without requirement to attend or respond to them, while an unrelated auditory task was performed), and lasted for 41 minutes. Results showed that detection performance after saturation deteriorated, but only for the stimulated hemifield. In addition, fMRI multivoxel pattern analysis (classifying the presented stimulus from BOLD activation patterns in identified stimulus-responsive areas) showed a drop in classification accuracy only for the saturated hemifield. This drop in MVPA classification performance was significant for activation patterns during the localizer scan, while a similar decline was not significant during the TDT task itself. Lastly, changes in BOLD activation were correlated with MVPA classifier performance, and with TDT behavioral performance, while subjective sleepiness was correlated with TDT behavioral performance.

I believe that this is a well-conceived and well-executed experiment that connects several inter-related, but theoretical perspectives that have so far existed rather isolated from each other. The TDT is a good choice of task for testing the hypothesis of passively induced neural fatigue, given its reliance on highly localized neural processing. The use of passive presentation to establish neural saturation is also an elegant way to avoid motivational factors usually intertwined with fatigue processes. From my reading, I believe that the data convincingly show that over-use of local neural pools can induce a) localized hypo-reactivity and degrade pattern representation, and B) performance decline confined to the saturated location. I mostly have some suggestions about the theoretical positioning of the paper.

We would like to thank the Reviewer for the careful review and for the positive comments.

There are two areas of research that have developed substantial thinking around the concept of (local) neural fatigue. The first is the area of local use-dependent sleep (Krueger et al. 2019; Siclari & Tononi 2017). The authors touch on this briefly in discussing the work by Vyazovzkiy et al. (2011). However, I believe that this area has produced a more elaborate body of work that may be of relevance to the current study. The idea that local neural ensembles temporarily go into an off-state due to over-use is highly similar to the concept of neural fatigue as advanced here. Both human (e.g. Huber et al. 2004) and animal work within this area, have inspired ideas about its mechanisms (e.g. neuronal sleep regulation through local inflammation), and functions (e.g. facilitating plasticity). In the latter aspect, this area also directly connects to the Texture Discrimination Task used here, which - as the authors indicate - had its origins perceptual learning, and sleep-dependent plasticity.

The second area that might be of relevance (esp in relation to the MVPA analysis) is the work on repetition suppression (i.e. reduced neural response to a repeated presentation of a stimulus). A prominent debate in this literature is whether repetition suppression reflects neural fatigue (an overall reduction in neural responsiveness to the stimulus), or neural sharpening (fine tuning of neural representational patterns). The current saturation paradigm could be seen as an extreme form of repetition suppression. Findings of reduced pattern classification in the current study might inform the debate on fatigue vs sharpening in the

area of repetition suppression (for reviews see Grill-Spector et al. 2006; Barron et al. 2016). Without forcing any obligation to veer too far away from the authors' original writing, I wonder if there is any merit to elaborating on (either or both) the above areas a bit more. I believe the current study would be in a good position to connect to these research areas, and both be informed by, and inform their respective theoretical debates.

We thank the Reviewer for the suggestion to broaden the theoretical scope of our work. We have integrated notions from the suggested literatures to the discussion section of the manuscript (p. 23-24).

Another comment about the theoretical framing is that the rationale is much built on contrasting the 'functional' and the 'motivational' accounts of fatigue. In the abstract they are presented as opposing views: "... it is unclear whether alterations due to prolonged mental effort take place in over-worked task-relevant neuronal assemblies, or in task unrelated networks involved in motivation regulation. Here, we tackle this question by means of a paradigm based on repeated, passive visual stimulation, ...". The introduction is a bit more cautious, saying that: "...the causes of mental fatigue remain elusive but have been theorized along a functional and a motivational axis". I follow the authors that the current design effectively isolates functional fatigue from motivational decline. At the same time, these effects need not be mutually exclusive, and could well occur simultaneously in most situations. I believe that a marked strength of the current study is that it probes neural fatigue, free motivational effects. However, I don't believe the study can arbitrate between functional and motivational accounts (as the abstract suggests).

We are sorry for the confusing wording of the abstract. We have removed the misleading sentence from the abstract and included further considerations on the topic in the discussion (p.27).

Specific comments

- Regarding the behavioral correlations, I couldn't quite make up what the TDT performance estimate represented (and how to interpret it). From the Methods Statistical analyses it was described as the random coefficients of the GLMM, but it wasn't clear to me which coefficient that was (is it the Session x Quadrant interaction?). I wonder if it is possible to describe the model in formula, and indicate which coefficient is used to run the correlations.

Indeed, the employed coefficients are from the significant interaction between Session and Quadrant. This was stated on line 290, page 14. Yet, as this was unclear to other Reviewers as well, we point it out again in line 375.

The formula has been included in the relevant section of the statistical analyses.

- Page 3 paragraph 3: references seem not formatted. "(Blain, Mednick, Gergelyfi)"

This issue has been fixed.

Reviewer #2 (Remarks to the Author):

The authors of the study titled ‘Neural fatigue by passive induction’ set out to resolve the question of whether cognitive fatigue occurs due to depletion of neural resources (functional account) or changes in motivation to pursue rewards (motivational account). Predicting that passive stimulation of neural circuits involved in effortful activity would induce fatigue, they present a paradigm where two blocks of perceptual decision-making task are interspersed with a block of prolonged exposition (‘saturation’) to perceptual stimulus in one of the visual fields targeted in the decision-making task and compare performance, patterns of brain activation in individually-calibrated ROIs, mean activation in the ROIs and subjective reports of fatigue and sleepiness. The results show a drop in performance, a decrease in MVPA classifier accuracy and a decrease in mean brain activation in clusters in V4 and V5 specific to trials and ROIs targeted by the saturation procedure. They find that the drop in mean brain activity (but not classifier performance) is correlated with the strength of effect on performance. The task is successful in eliciting higher levels of fatigue and sleepiness as measured with self-report questionnaires, but only the sleepiness increase is shown to correlate with a drop in performance. Authors conclude that passive stimulation of brain circuits induces neural fatigue specific to these circuits, mirrored in a drop in performance only in the task recruiting those and that these changes are responsible for the mental fatigue.

This is timely topic and an interesting study using a novel methodology to examine the still under-researched phenomenon of fatigue. Including links to the data and analysis scripts must be appreciated and commended. Although there are elements of the manuscript that are real strengths, the work does not provide compelling evidence in favour of their conclusions. Firstly, the initial theoretical tension (between motivation and neuronal fatigue accounts) as being opposed is over-stated, the task cannot rule between such accounts, the sample size is relatively small, and many statistical results are not convincing. For these reasons the manuscript falls somewhat short of conclusively showing neuronal fatigue in task-sensitive circuits is responsible for mental fatigue, which may leave it unable to make a significant impact on the field.

Major:

1. The main premise of the study is that accounts of fatigue either argue it is purely a motivational phenomena and other accounts suggesting it is based on neuronal fatigue. However, these accounts are not mutually exclusive and the dichotomy is somewhat overstated. More recent accounts all suggest that increases in neuronal fatigue might influence motivation through processes similar to meta-cognition (see Muller & Apps, 2019, the work of Steve Fleming or Meyniel et al., 2013). These accounts argue that costs accumulate in task-processing areas, and are monitored in other systems that then influence subsequent valuations of whether to keep working at the task (Pessiglione et al., 2018, Brain; Muller et al., 2021, Nature Comms). In this sense, task performance can depend on both processes. I.e. two people could have similar levels of neuronal fatigue but one person suffers from poor performance and another doesn’t, because they can compensate by being more motivated. The problem for this manuscript, is that the whole paper is framed around showing effects are related to neuronal fatigue not motivation, but the second theory is never ruled out. As a result, much of the framing of the paper is over-stated and misleading.

We would like to thank the Reviewer for their thoughtful remarks. We no longer mention in the abstract that we tackle the question of functional versus motivational accounts of fatigue, and we have included a less univocal interpretation of our results in the discussion. It was never our

intention to claim that we excluded the implication of motivational processes in fatigue, as this issue is obviously not addressed in our experiment. Rather, we claim to provide evidence that fatigue can be observed even when motivational influences have been ruled out.

2. Linked to this, conceptually, there are some issues with referring to performance drop as ‘objective fatigue’. Fatigue is a complex and multi-faceted phenomenon, often accompanied by loss of motivation and a drop in performance, however, these objective measures are better seen as potential behavioural correlates of fatigue, which is an inherently subjective phenomenon, rather than its defining features. There are a number of problems with the definitions used throughout. The most problematic being that the study does not manage to show a relationship between the subjective measures of fatigue and behavioural and neural measures. As a result, much of the introduction and discussion about this work being about “fatigue” is misleading. Moreover, they only link task measures to subjective sleepiness and not subjective fatigue. This is a major problem for the framing of the paper is all around mental fatigue.

Fatigue is defined as being composed of both objective and subjective components and we must respectfully disagree with the view proposed by the Reviewer that the subjective component would be a better, more fundamental index of fatigue. The choice of favoring subjective over objective aspects depends on the particular question that the experiment addresses.

The lack of correlation between objective and subjective components of fatigue is the norm rather than the exception in the literature, in contrast to what the Reviewer seems to suggest, strengthening the point that fatigue is really multidimensional. In fact, we found such a correlation in our previous paper that has been recently accepted for publication (Ioannucci et al., 2022), and while it failed to reach significance in the present experiment, we found a significant correlation with sleepiness, a concept which is very close to fatigue. As we argue in the Discussion, we believe that the supine position of the participants and the long duration of the experiment may have favoured this particular relationship.

Relatedly: a. Multidimensional Fatigue Inventory (once erroneously referred to as “Index” (p.13, please correct) was developed as an assessment of a persistent, pathological symptom of fatigue (developed specifically in cancer patients) rather than an evaluation of (non-pathological) fatigue that develops in response to exertion. With items like “I think I do a lot in a day” or “I tire easily” this inventory is not particularly suited to detect changes in effort-induced fatigue levels. I do not blame authors for this as a suitable measure has not, to my knowledge, been devised, however, it does lead to my questioning the justifiability of interpreting their results in terms of mental fatigue.

We agree with the Reviewer’s assessment and have in fact used a modified version of the MFI that does not include these questions (only the General Fatigue and Mental Fatigue questions are included). We have used this modified version of the MFI in previous studies and were quite satisfied with the reliability and validity of the results (Ioannucci et al., 2022; Gergelyfi et al., 2015; Benoit et al., 2019; Gergelyfi et al., 2021). We apologise for failing to report this detail in the earlier version of the manuscript. This is now mentioned in the Methods section (p.15).

b. Following up from b), Multidimensional Fatigue Inventory is, by name and nature, multidimensional where authors purport to measure mental fatigue specifically

Similarly, that is why we have used the modified MFI, which includes only the General Fatigue and Mental fatigue questions.

c. Finally, the authors compare the subjective measure of fatigue change to ‘objective measure’ by the means of correlating change in MFI score with the participant-specific coefficient in GLMM predicting the accuracy in the task. As far as I understand this is a coefficient quantifying interaction between task session (pre- vs post-passive stimulation session) and visual field (or ‘Quadrant’: saturated vs non-saturated), so the potential experimental manipulations that could explain the drop in performance. To fully explore this relationship it would be advisable to examine whether subjective fatigue is related to a drop in performance between sessions in and of itself, regardless of the saturation procedure. Although there was no significant difference between accuracy in two experimental sessions in non-saturated quadrant, based on the accuracy plot (Figure 5) and specifically the error bar for session 2 - non-saturated quadrant it appears that there were more substantial individual differences in participants’ drop in performance in this condition.

We have ran the suggested correlation and the result is not significant ($R_{(22)} = -0.27$, $p = 0.199$). We provide this result specifically for the benefit of the Reviewer, as this was not mentioned in the pre-registered design.

3. During the saturation session participants were performing cognitively demanding auditory tasks. Very little is explained regarding the nature of these tasks, e.g., whether the effort level of these tasks was individually adjusted. It is likely that these tasks also had an effect on any changes in fatigue levels, so more information on these tasks would be welcome and on data on what participants were doing. This seems crucial to understand what “saturation” and fatigue effects actually mean in this task.

We apologise for this lack of detail. The auditory tasks are now described in detail in the relevant section of the paper (p.9).

4. Overall, many of the key results that one would expect are not present (see links to subjective fatigue above), some results that are borderline or non-significant but interpreted as significant (see Fig.3c), the between-subject variability seems high in all key figures and in some analyses classifier accuracy is very close to 50% in all conditions. Given the small sample size, this leaves a concern that many of the results are not robust, and will likely not replicate. Of particular concern is that several analyses rely on correlations, which are known in small samples to be inflated (Yarkoni et al., 2009). Overall, even though I am sympathetic to the notion that interesting studies don’t need large samples, this particular pattern of statistical results in a small sample appears concerning.

It is unclear what the Reviewer means by “key results that one would expect”. From comment #2 above, we suppose that she/he refers to the correlation between objective and subjective fatigue. However, as mentioned above, based on the literature, one would not expect to reliably find a relationship between subjective and objective fatigue (DeLuca, 2007).

Regarding figure 3c, the excerpt describing it states: “No significant effects were found when looking at the MVPA results in the TDT task, though the interaction of Quadrant and Session followed a downwards trend exclusively for the saturated condition ($F_{(1, 23)} = 3.24$, $p = .085$, $\eta^2_p = 0.124$; see figure 3c).”. We thus did not interpret it as significant at all, but presented it for the sake of completeness and consistency with pre-registered design.

Concerning the replicability of the findings from a study, they may be judged based on the effect sizes, which are clearly reported in the manuscript. Besides, we must point out to the Reviewer that

several of the reported results are replications themselves of previous work (Ioannucci 2022), which is how we determined the appropriate sample size for the work being reviewed.

Additionally, a sample size of 24 subjects is in line with the average in fMRI research, including between-subjects research (Szucs & Ioannidis, 2020). As our design is within-subjects, known to have greater statistical power, we fail to see according to which standard the present sample may be considered “small”. Additionally, too large sample sizes also have their drawbacks, as they may highlight as significant phenomena that have limited practical interest (Wilson et al., 2022).

5. Linked to that point, is the fact that figure 5, appears to have 2 outliers in the perceived sleepiness scores top left and bottom right of the plot). Excluding those and reanalysing the data would be advisable to show whether or not these are driving the significance of the correlation.

Even though there is no statistical principle on which to base the exclusion of these datapoints (which are well within the 3 SD range), we have ran the suggested analysis and found that the Pearson correlation between sleepiness scores and TDT performance became indeed non-significant ($R_{(19)} = -0.42$, $p = 0.0621$). However, a better way to assess robustness of the results to outliers is to run the same analysis with Spearman (rank) correlation. This analysis confirmed the correlation ($R_{(21)} = -.52$, $p=0.0099$). We have included the latter result in our manuscript (p.21).

As a side note, if we were to adopt the same subjective criterion for excluding “outlier” datapoints in the correlation between MFI scores and TDT performance, then it would become significant (in the expected direction) by dropping a single datapoint.

6. In general, many aspects of the manuscript were a little unclear and lacking in important details (see each of the minor comments below). In particular, what tests were being conducted and how statistical results were derived was hard to follow and lacking the clarity required for reproducibility. Figure legends were lacking the detail required to understand what was being plotted

As requested also by Reviewer 3, we have added the rationale for each statistical test and hope this provides the required clarity. Unfortunately, the Reviewer’s comment is not detailed enough to allow us to determine which figure legend needs updating.

Minor:

7. Overall there is a general lack of reference to the most recent develops for research in mental fatigue. The only papers referenced after 2016 are to methodologically relevant papers, not conceptual or empirical on the topic at hand – fatigue.

We must point out that the assertion that works on fatigue dating after 2016 were not included is factually wrong (Benoit et al., 2019; Gergelyfi et al., 2021; Tran et al., 2020). Nevertheless, we are receptive to the Reviewer’s useful suggestion to broaden the scope of our work and include more recent research, and have done so (Müller et al., 2021; Müller & Apps, 2019; Wiehler et al., 2022).

8. The title could elaborate a little more on the contents of the study for greater clarity and impact. Passive induction is not a well parsed phrase or clear to people even in the field

We must respectfully disagree with the point raised by the Reviewer, as we believe the chosen title to be appropriate.

9. As authors point out (in discussion, p.25), the approach of linking (or attempting to link) behavioural measures and MVPA classifier accuracy is not used often. It would therefore be useful to include a more elaborate rationale for this choice and underlying assumptions.

We now provide a more elaborate rationale when introducing this analysis (p.14).

10. Please include the correlation methods used

The specification of the employed correlation methods (Pearson, Spearman) have been added in the Statistical Analysis section (p.14-15).

11. In figure 2 please describe what the error bars depict

This has been added to Figure 2 legends.

12. Since the results of staircase procedure were capped, please include information how many participants were capped at the maximum 0.6 ISI and how this affected their performance

None of the participant reached the cap of 0.6 test-day ISI, with the highest being 0.51. This is now mentioned in the manuscript (p.16).

13. Word-use on page 10 ('betas') and page 12 ('brains') should be more precise

We are unsure as to which issue the Reviewer is pointing out here. However, we have changed the usage of "betas" to "beta coefficients" and "brains" to "brain images". We hope this addresses the Reviewer's concern satisfactorily.

14. Page 23: "Indeed, V4 is thought to contain neuronal populations tuned to the orientation of basic visual stimuli (Roe et al., 2012), such as the lines employed in the present study. While V5 is believed to be implicated, to some degree, in the perception of movement (Born & Bradley, 2005), therefore reasonably being expected to respond to the stimuli of the localizer trials, due to their continuous 7.5 Hz fluttering." - please correct the grammar

We have modified these sentences to improve their readability (p.26):

"Indeed, V4 is thought to contain neuronal populations tuned to the orientation of basic visual stimuli (Roe et al., 2012), such as the lines employed in the present study, while V5 is believed to be involved in motion perception (Born & Bradley, 2005), making it presumably sensitive to the continuous 7.5 Hz fluttering of the visual stimuli."

15. Page 25: "These systems were identified as the "limbic-basal ganglia-thalamus-frontal" network in the case of the facilitation system, and as the "insular-posterior cingulate" in the case of the inhibition system, with the latter found to be involved in fatigue on top of task-related regions only for pathological samples." - please include a reference

We have added the citation of the relevant paper at the end of each assertion concerning it (p.30-31).

16. Page 25: "At any rate, the evidence supporting the existence of these systems is limited and comes from studies that do not involve perceptual tasks such as the present one." - please include references

This relates again to the same reference, which has been repeated in the corresponding location in the manuscript (p.31).

17. Please ensure that acronyms (MVPA, TDT) are all capitalised in the subheadings

This has been implemented.

References:

- Benoit, C.-E., Solopchuk, O., Borragán, G., Carbonnelle, A., Van Durme, S., & Zénon, A. (2019). Cognitive task avoidance correlates with fatigue-induced performance decrement but not with subjective fatigue. *Neuropsychologia*, *123*, 30–40.
<https://doi.org/10.1016/j.neuropsychologia.2018.06.017>
- DeLuca, J. (2007). *Fatigue as a Window to the Brain*. MIT Press.
- Gergelyfi, M., Sanz-Arigita, E. J., Solopchuk, O., Dricot, L., Jacob, B., & Zénon, A. (2021). Mental fatigue correlates with depression of task-related network and augmented DMN activity but spares the reward circuit. *NeuroImage*, *243*, 118532.
<https://doi.org/10.1016/j.neuroimage.2021.118532>
- Ioannucci, S., Borragán, G., & Zénon, A. (2022). Passive visual stimulation induces fatigue under conditions of high arousal elicited by auditory tasks. *Journal of Experimental Psychology: General*. Advance online publication. <https://doi.org/10.1037/xge0001224>
- Müller, T., & Apps, M. A. J. (2019). Motivational fatigue: A neurocognitive framework for the impact of effortful exertion on subsequent motivation. *Neuropsychologia*, *123*, 141–151.
<https://doi.org/10.1016/j.neuropsychologia.2018.04.030>
- Müller, T., Klein-Flügge, M. C., Manohar, S. G., Husain, M., & Apps, M. A. J. (2021). Neural and computational mechanisms of momentary fatigue and persistence in effort-based choice. *Nature Communications*, *12*(1), 4593. <https://doi.org/10.1038/s41467-021-24927-7>
- Szucs, D., & Ioannidis, J. P. (2020). Sample size evolution in neuroimaging research: An evaluation of highly-cited studies (1990-2012) and of latest practices (2017-2018) in high-impact journals. *NeuroImage*, *221*, 117164. <https://doi.org/10.1016/j.neuroimage.2020.117164>
- Tran, Y., Craig, A., Craig, R., Chai, R., & Nguyen, H. (2020). The influence of mental fatigue on brain activity: Evidence from a systematic review with meta-analyses. *Psychophysiology*, *57*(5), e13554. <https://doi.org/10.1111/psyp.13554>
- Wiehler, A., Branzoli, F., Adanyeguh, I., Mochel, F., & Pessiglione, M. (2022). A neuro-metabolic account of why daylong cognitive work alters the control of economic decisions. *Current Biology*, *0*(0). <https://doi.org/10.1016/j.cub.2022.07.010>
- Wilson, B. M., Harris, C. R., & Wixted, J. T. (2022). Theoretical false positive psychology. *Psychonomic Bulletin & Review*. <https://doi.org/10.3758/s13423-022-02098-w>

Reviewer #3 (Remarks to the Author):

In this manuscript, Ioannucci et al. used passive visual stimulation of a specific region of the visual field to induce a loss of performance specific to that region. They then used fMRI to find brain correlates of this effect, showing functional brain alterations in task-relevant networks. All in all, the paper is well-written, the experiment is cleverly devised and the results sound convincing albeit intriguing. I only have a couple of minor comments.

My main concern is about the interpretation of the results regarding the subjective feeling of (cognitive) fatigue. The beauty of the design is about the specificity of the effect. However, fatigue is a general feeling that seems to transfer from one task to the other (after a long session of reading/writing, I'm too tired to play chess!). Thus, even if I'm convinced by the results, I would be reluctant to call "cognitive fatigue" a decrease of performance specific to a very specific task. On a related topic, I would like the authors to elaborate on how they would imagine the link between (local) brain fatigue and what is reported by participants when they are required to rate their (global) level of fatigue. Is it supposed to be an average of the whole brain? Or maybe the degree of fatigue of the most "exhausted" region? Finally, my intuitive assumption would have been that vision (and its correlates) might be the most difficult function to fatigue... After all, we are receiving visual information all day long on the whole visual field every day... On the contrary, some functions (e.g. executive ones) seem particularly susceptible to fatigue. I understand why the authors chose this task given the methods they used, but as the paper is framed about cognitive fatigue, I believe they should elaborate about how their results could be extended to the cognitive fatigue experienced in everyday life.

We thank the Reviewer for his appreciative remarks. We have enlarged the scope of our discussion to address his comments on the topic of cognitive fatigue, with examples from everyday life. We would like to point out also that in previous work based on the same protocol, which was recently accepted for publication (Ioannucci et al., 2022), we have found direct correlations between local objective fatigue and global subjective fatigue, which indicates that there is some proportional relationship between the two.

The rationale of each analyze and expected results could be more explicitly detailed in the result section to facilitate the reading. This is particularly the case for negative results.

The rationale behind each analysis has been added in the statistical analysis section (p.13-15).

I'm not sure to understand the number of participants. I understand that one of the participants was excluded from fMRI analyses (and kept in behavioral ones) because of artefacts. However, since the most important analyses involve baseline and conclusion sessions, I'm not sure it makes sense to keep the participant that failed to respond in the concluding session...

We apologise for not making this clearer in the paper. In fact the participant was excluded due to the system not registering the button presses properly, rather than due to a lack of response from the participant. That is why we kept the data for the neuroimaging analysis. We have corrected this in the text.

Univariate analyses: By construction, the two clusters identified (Fig 4a) must be within the overlapping ROIs (Fig 3a) but it seems that at least the bigger one is not in a region common to most subjects. However, it is quite difficult to assess from eyeballing, would it be possible to provide an overlap of these two maps (flipping ROI when required).

The two clusters are within the overlapping ROI, as their union was employed as mask in the univariate analysis. In the attached image we can see the flipped LQ joint ROI in green, along with the RQ joint ROI in blue, and the larger cluster from the univariate analysis laying at their intersection, in red.

The larger cluster appears to be less shared between participants, probably due to its size, since in such a large brain patch, fewer participants had activations in exactly the same place. This did not translate to fewer participants showing significant activity in that portion of space, especially when considering reversal.

Voxel-wise baseline activity in response to stimuli & change in classifying accuracy : If I'm correct, the voxels that exhibited the lowest beta estimates at baseline in response to stimuli in the non-saturated quadrant displayed the largest loss (opposite of what was observed in the saturated one). I was quite surprised by this result (I would have expected a lack of relation). Could the authors elaborate on this finding?

We were also initially surprised by this finding. However, we must point out that the post-hoc analyses restricted to either the non-saturated or saturated quadrants are not significant ($p=0.69$ and $p=0.51$, respectively). Therefore, the only result we can interpret is the interaction between baseline activation and saturated quadrant, as reported in the manuscript.

Fig 4.C (axe and legend): I'm not sure speaking about "neurons" is appropriate here (speaking about the brain activity of neurons was quite funny though!)

We apologize for this grammar error and thank the Reviewer for pointing it out.

Which random coefficients of the GLMM were correlated with questionnaire scores?

All the Reviewers raised this issue and we apologize for this lack of clarity. The employed coefficients are the ones for the interaction between Quadrant and Session. This was stated on page 14 line 290, and we have added an ulterior mention of this on line 375.

According to the result section, I believe sleepiness was correlated to change of performance in the saturated quadrant but not in the non-saturated one?

The coefficients used here indicate the difference in the evolution of performance between the saturated and non saturated quadrants. A negative coefficient indicates a larger loss of performance between baseline and conclusion in the saturated than in the non-saturated quadrant. This has been clarified in the text (p.22).

Was there some correlation between questionnaires and fMRI findings?

Correlation between the fatigue questionnaire scores and the change in estimated brain activity in the saturated ROIs was not found to be statistically significant ($R_{(22)} = -0.15$, $p = 0.49$). Similar results were found for the correlation between change in estimated brain activity in the saturated ROIs and the sleepiness scale scores ($R_{(21)} = -0.29$, $p = 0.17$).

References:

Ioannucci, S., Borragán, G., & Zénon, A. (2022). Passive visual stimulation induces fatigue under conditions of high arousal elicited by auditory tasks. *Journal of Experimental Psychology: General*. Advance online publication. <https://doi.org/10.1037/xge0001224>

Reviewers' comments:

Reviewer #1 (Remarks to the Author):

I'd like to thank the authors for their revision and thoughtful response to the Reviewers' comments. I feel all points I've raised have been adequately clarified. I have only one comment after reading the revised Discussion, where the authors speculate about the occurrence neural (visual) in real-world scenarios (eg in office work). I could imagine that we don't often encounter effects of such visual fatigue in everyday life, because strict ocular fixation is extremely rare for longer periods of time. Eye movements and micro saccades would likely prevent such saturation to reach noticeable levels. To the point of Reviewer3, the visual system is equipped to deal with a very high throughput of sensory information, and the visual/ocular system likely has sophisticated mechanisms in place to allow for this without too strong fatigue effects.

Reviewer #2 (Remarks to the Author):

I thank the authors for replying to my previous comments. Although I acknowledge a number of the points, I still do not believe that the manuscript provides strong evidence for the conclusions and do not feel robust rebuttals were provided to several of my comments. For these reasons my initial reservations regarding the manuscript remain.

1. Although authors note that their sample size is average within the fMRI literature, this includes a large proportion of small n studies from a number of years ago, and a number of research groups have noted the insufficiency of such samples. The average of most ongoing research is higher. This is therefore not a strong argument for why n=24 is a robust sample size.

2. Once again, several of the reported results are correlations (Figure 4c and Figure 5). These are very small samples to conduct such correlation analyses reliably (Bonnett & Wright 2000). These are used to draw significant inferences upon and the basis of conclusions of the manuscript.

3.. The authors continue to interpret results that aren't significant as meaningful. line 348 "though the interaction of Quadrant and Session followed a downwards trend exclusively for the saturated condition ($F(1, 23) = 3.24$, $p = .085$, $\eta^2p = 0.124$; see figure 3c)," despite claiming in the rebuttal that this was not the case.

4. Not including results that weren't in the pre-registration. The authors note that they have provided results to the reviewer but not included in the manuscript "because they were not in the pre-registration", but that is not a sound justification. This isn't a registered report and reviewers did not get to indicate they felt that the pre-registration was exhaustive. Thus, analyses that are proposed by reviewers that are justifiable, in this case which relate to comments made by more than one reviewer, can be included in the manuscript.

Reviewer #3 (Remarks to the Author):

The authors carefully addressed all my concerns. I would like to congratulate them again for a very nice paper.

Reviewer #1 (Remarks to the Author):

I'd like to thank the authors for their revision and thoughtful response to the Reviewers' comments. I feel all points I've raised have been adequately clarified. I have only one comment after reading the revised Discussion, where the authors speculate about the occurrence neural (visual) in real-world scenarios (eg in office work). I could imagine that we don't often encounter effects of such visual fatigue in everyday life, because strict ocular fixation is extremely rare for longer periods of time. Eye movements and micro saccades would likely prevent such saturation to reach noticeable levels. To the point of Reviewer3, the visual system is equipped to deal with a very high throughput of sensory information, and the visual/ocular system likely has sophisticated mechanisms in place to allow for this without too strong fatigue effects.

We thank the Reviewer for this insightful remark and have added this argument to the Discussion of our manuscript (page 24).

Reviewer #2 (Remarks to the Author):

I thank the authors for replying to my previous comments. Although I acknowledge a number of the points, I still do not believe that the manuscript provides strong evidence for the conclusions and do not feel robust rebuttals were provided to several of my comments. For these reasons my initial reservations regarding the manuscript remain.

1. Although authors note that their sample size is average within the fMRI literature, this includes a large proportion of small n studies from a number of years ago, and a number of research groups have noted the insufficiency of such samples. The average of most ongoing research is higher. This is therefore not a strong argument for why n=24 is a robust sample size.

We thank the Reviewer for this remark. Indeed, there is a desirable trend of increase in the sample sizes in the fMRI literature, with n=29 being the median sample size as of 2017, for both between and within subjects designs (Poldrack et al., 2017).

We must respectfully point out that we never intended to make the general claim that “n=24 is a robust sample size” but rather that it is an adequate sample size for our specific experiment, in line with contemporary fMRI standards. This claim rests upon these, so far directly undisputed, premises:

- We reported the behavioural effects in 3 previous separate, similar or smaller, samples (N = 24 / 16 / 14) (Ioannucci et al., 2022).
- We use a within-subjects design, which has stronger power than between-subjects. So our sample size, albeit on the lower side of the median fMRI current standards, would still achieve larger statistical power than some between-subjects studies involving more participants.
- We are assessing strongly localized changes of brain activity in reaction to visual stimulation, which are known to display the highest effect sizes in terms of BOLD signal change with respect to changes in higher order cognitive processes with complicated designs (Cremers et al., 2017; Poldrack et al., 2017).
- In one of the papers cited to question our sample size choice (Poldrack et al., 2017), pre-registration, functional localizers, multivariate analyses and non-parametric permutation tests (see also (Durnez et al., 2014)) are indicated among the possible solutions to alleviate the endemic issue of low statistical power in fMRI literature. All of these solutions were employed in our work precisely to follow the current best practices in the field.

Moreover, the utility of post-hoc power analyses is heavily controversial, see for example:

“Power analysis is an indispensable component of planning clinical research studies. However, when used to indicate power for outcomes already observed, it is not only conceptually flawed but also analytically misleading. Our simulation results show that such power analyses do not indicate true power for detecting statistical significance, since post hoc power estimates are generally variable in the range of practical interest and can be very different from the true power.” (Zhang et al., 2019)

Nevertheless, to address the Reviewer’s point and as requested by the Editors, we have ran a post-hoc power analysis on the reported fMRI results and included the following text on page 19:

We have run exploratory tests to evaluate the statistical power of our reported results. Concerning the repeated measures ANOVA on the MVPA classifier accuracies, the results from G*Power (Faul et al., 2007) state that with our reported effect size and employed sample size, a statistical power of 90% was reached, indicating that a significant result could have been found with a sample of only 10 participants.

With respect to the univariate analysis, we calculated the average Cohen’s d measure of effect size in the significant clusters for the post – pre contrast, which yielded a value of $d = -0.52$. This is generally interpreted as a medium effect size (Fritz et al., 2012), and matches the higher end of the distribution of fMRI effect sizes reported in the study on the topic from Poldrack and colleagues (Poldrack et al., 2017). With our sample size, this translates in an achieved statistical power level of 68%, as calculated with G*Power. Although reporting effect sizes should be standard practice in scientific endeavors involving statistical analyses (Lakens, 2013), post-hoc power analyses should be approached with caution, as their use is a source of much debate within the scholarly community (Zhang et al., 2019).

2. Once again, several of the reported results are correlations (Figure 4c and Figure 5). These are very small samples to conduct such correlation analyses reliably (Bonnett & Wright 2000). These are used to draw significant inferences upon and the basis of conclusions of the manuscript.

We have added the suggested citation and mention this limitation in the relevant section of the manuscript (page 26):

Another point to be raised here is that beta-behavior correlation analyses in fMRI (Lebreton et al., 2019), and in general in the presence of small sample sizes (Bonnett & Wright, 2000) should be interpreted with caution. Replications will be necessary to conclusively validate the correlations highlighted in the present work.

3. The authors continue to interpret results that aren't significant as meaningful. line 348 "though the interaction of Quadrant and Session followed a downwards trend exclusively for the saturated condition ($F(1, 23) = 3.24$, $p = .085$, $\eta^2_p = 0.124$; see figure 3c)," despite claiming in the rebuttal that this was not the case.

We have replaced this text with (page 14):

Concerning the assessment of the MVPA results employing brain activity during the TDT, the interaction between Quadrant and Session failed to reach statistical significance ($F(1, 23) = 3.24$, $p = .085$, $\eta^2_p = 0.124$; see figure 3c), along with any other effect.

4. Not including results that weren't in the pre-registration. The authors note that they have provided results to the reviewer but not included in the manuscript "because they were not in the pre-registration", but that is not a sound justification. This isn't a registered report and

reviewers did not get to indicate they felt that the pre-registration was exhaustive. Thus, analyses that are proposed by reviewers that are justifiable, in this case which relate to comments made by more than one reviewer, can be included in the manuscript.

We have included the analysis in the text (page 18), as requested:

No significant correlation was found between the performance change in the TDT and the evolution of subjective fatigue between sessions, whether employing the subject coefficients of the mixed model for the interaction between session and quadrant ($R_{(22)} = -0.22$, $CI = -0.57, 0.22$, $p = 0.29$) or solely that of session ($R_{(22)} = -0.30$, $CI = -0.63, 0.11$, $p = 0.15$).

References

- Bonett, D. G., & Wright, T. A. (2000). Sample size requirements for estimating Pearson, Kendall and Spearman correlations. *Psychometrika*, *65*, 23–28. <https://doi.org/10.1007/BF02294183>
- Cremers, H. R., Wager, T. D., & Yarkoni, T. (2017). The relation between statistical power and inference in fMRI. *PLOS ONE*, *12*(11), e0184923. <https://doi.org/10.1371/journal.pone.0184923>
- Durnez, J., Moerkerke, B., & Nichols, T. E. (2014). Post-hoc power estimation for topological inference in fMRI. *NeuroImage*, *84*, 45–64. <https://doi.org/10.1016/j.neuroimage.2013.07.072>
- Faul, F., Erdfelder, E., Lang, A.-G., & Buchner, A. (2007). G*Power 3: A flexible statistical power analysis program for the social, behavioral, and biomedical sciences. *Behavior Research Methods*, *39*, 175–191. <https://doi.org/10.3758/BF03193146>
- Fritz, C. O., Morris, P. E., & Richler, J. J. (2012). Effect size estimates: Current use, calculations, and interpretation. *Journal of Experimental Psychology: General*, *141*, 2–18. <https://doi.org/10.1037/a0024338>
- Ioannucci, S., Borragán, G., & Zénon, A. (2022). Passive visual stimulation induces fatigue under conditions of high arousal elicited by auditory tasks. *Journal of Experimental Psychology: General*, *151*(12), 3097–3113. <https://doi.org/10.1037/xge0001224>
- Lakens, D. (2013). Calculating and reporting effect sizes to facilitate cumulative science: A practical primer for t-tests and ANOVAs. *Frontiers in Psychology*, *4*. <https://www.frontiersin.org/articles/10.3389/fpsyg.2013.00863>
- Lebreton, M., Bavard, S., Daunizeau, J., & Palminteri, S. (2019). Assessing inter-individual differences with task-related functional neuroimaging. *Nature Human Behaviour*, *3*(9), Article 9. <https://doi.org/10.1038/s41562-019-0681-8>
- Poldrack, R. A., Baker, C. I., Durnez, J., Gorgolewski, K. J., Matthews, P. M., Munafò, M. R., Nichols, T. E., Poline, J.-B., Vul, E., & Yarkoni, T. (2017). Scanning the horizon: Towards transparent and reproducible neuroimaging research. *Nature Reviews Neuroscience*, *18*(2), Article 2. <https://doi.org/10.1038/nrn.2016.167>
- Zhang, Y., Hedo, R., Rivera, A., Rull, R., Richardson, S., & Tu, X. M. (2019). Post hoc power analysis: Is it an informative and meaningful analysis? *General Psychiatry*, *32*(4), e100069. <https://doi.org/10.1136/gpsych-2019-100069>